

# Understanding the seasonality and climatology of aerosols in Africa through evaluation of CCAM aerosol simulations against AERONET measurements

Hannah M. Horowitz[1], Rebecca M. Garland[2,3], Marcus Thatcher[4], Willem A. Landman[2,5], Zane Dedekind[2], Jacobus van der Merwe[2], Francois A. Engelbrecht[2,6]

[1]Department of Earth & Planetary Sciences, Harvard University, Cambridge, MA, 02138, USA
[2]Natural Resources and the Environment Unit, Council for Scientific and Industrial Research, Pretoria, 0001, South Africa
[3]Climatology Research Group, North West University, Potchefstroom, 2520, South Africa
[4]Marine and Atmospheric Research, Commonwealth Scientific and Industrial Research Organisation, Melbourne, 3195, Australia
[5]Department of Geography, Geoinformatics and Meteorology, University of Pretoria, Hatfield, 0028, South Africa
[6]School of Geography, Archaeology and Environmental Studies, University of the Witwatersrand, Johannesburg, 2000, South Africa

*Correspondence to*: hmhorow@post.harvard.edu

**Abstract.** The sensitivity of climate models to the characterization of African aerosol particles is poorly understood. Africa is a major source of dust and biomass burning aerosols and so this represents an important research gap in understanding the impact of aerosols on radiative forcing of the climate system. Here we evaluate the current representation of aerosol particles in the Conformal Cubic Atmospheric Model (CCAM) with ground-based observations across Africa, and additionally provide an analysis of aerosol optical depth at 550 nm ($AOD_{550nm}$) and Ångström exponent data from thirty-four Aerosol Robotic Network (AERONET) sites.

Analysis of the 34 long-term AERONET sites confirms the importance of dust and biomass burning emissions to the seasonal cycle and magnitude of $AOD_{550nm}$ across the continent and the transport of these emissions to regions outside of the continent. Western African sites had the largest $AOD_{550nm}$ values, on average, with the timing and magnitude of $AOD_{550nm}$ maxima dominated by desert dust. The impact of dust on aerosol loading is also apparent at northern African sites, with peak $AOD_{550nm}$ occurring later than the western sites. The seasonal variation in the location of the intertropical convergence zone and associated northward shift in dust transport may be responsible for the shift in timing of maximum $AOD_{550nm}$ between the western and northern African sites. Southern African sites have the lowest $AOD_{550nm}$ values on average, and peak during the biomass burning period. The outflow of these aerosol particles was observed at Ascension Island and Reunion Island AERONET stations.

In general, CCAM captures well the seasonality of the AERONET data across the continent. The magnitude of modeled and observed multi-year monthly average $AOD_{550nm}$ overlap within ± 1 standard deviation of each other for at least 7 months at all sites except Reunion Island. The timing of peak $AOD_{550nm}$ in southern Africa in the model occurs one month prior to the observed peak, which does not align with the timing of maximum fire counts in the region. For the western and northern


African sites, it is evident that CCAM currently overestimates dust in some regions while others (e.g., the Arabian Peninsula) are better characterized. This may be due to overestimated dust lifetime, or that the characterization of the soil for these areas needs to be updated with local information. The CCAM simulated $AOD_{550nm}$ for the global domain is within the spread of previously published results from CMIP5 and AeroCom experiments for black carbon, organic carbon and sulfate aerosols.

The model's performance provides confidence for using the model to estimate large-scale regional impacts of African aerosols on radiative forcing, but local feedbacks between dust aerosols and climate over northern Africa and the Mediterranean may be overestimated.

## 1 Introduction

Africa contains the largest single sources of biomass burning emissions and dust globally (Crutzen and Andreae, 1990;

van der Werf et al., 2010; Schütz et al., 1981; Prospero et al., 2002). Dust aerosols, along with carbonaceous aerosols produced from biomass burning, are known to impact climate through direct scattering and absorption of radiation, and indirectly through their effects on cloud formation and properties. Black carbon is estimated to be second only to $CO_2$ in contributing to warming globally (Bond et al., 2013). Currently, the largest uncertainty in climate models is the impact of aerosols on the radiative balance of the Earth (Boucher et al., 2013).

Mineral dust emitted into the atmosphere primarily originates in topographic depressions (Prospero et al., 2002), consistent with the acceleration of winds in between mountains and plateaus (Evan et al., 2016). Meteorology plays a key role in the seasonality of dust emissions and transport in Africa. Latitudinal changes in the large-scale circulation, including the intertropical convergence zone (ITCZ) and the African monsoon, shift the location of maximum dust activity and transport of dust northward (~5˚N to ~20˚N) from winter through summer (Jankowiak and Tanre, 1992; Moulin et al., 1997;

Prospero et al., 2002; Schepanski et al., 2009; Leon et al., 2009). The movement of the ITCZ also determines the seasonality of precipitation, and so determines the onset and severity of dry season biomass burning in Africa. Most fires in Africa are set by humans during the dry season for agricultural practices, when there is a near absence of convection and lightning (e.g., Swap et al., 2003; Archibald, 2016). Maximum biomass burning activity thus shifts from June–September in southern Africa, to December–February in sub-Sahelian northern Africa (Haywood et al., 2008; Duncan et al., 2003; Cooke et al.,

1996). The magnitude of emissions in a given biomass burning season is largely determined by the amount of rainfall preceding burning (which is affected by climate variability such as the El Niño Southern Oscillation), as this affects the amount of vegetation that grows and can be burned (Swap et al., 2003; Anyamba et al., 2003; van der Werf et al., 2004). Biomass burning emissions in southern Africa contribute greatly to the region's aerosol burden and in many places dominate the seasonal cycle of the aerosol column in the region (Tesfaye et al., 2011; Queface et al., 2011; Sivakumar et al., 2010; Eck

et al., 2003), which in turn can have a significant impact on the regional climate (Abel et al., 2005; Winkler et al., 2008; Tummon et al., 2010). Although these two sources dominate total column aerosol in Africa, fine anthropogenic aerosols are





also observed, including at sites in the Sahara desert and off the coast of northern Africa (Rodriguez et al., 2011; Guirado et al., 2014).

In addition to the local and regional impacts of African dust and biomass burning aerosols near emission sources, the aerosol particles can also be transported long distances to impact other regions. Saharan dust is exported over the Atlantic Ocean, cooling the tropical North Atlantic and influencing Atlantic climate variability (Evan et al., 2011; Doherty and Evan, 2014). Climate change may reduce future dust emissions, thus leading to a positive warming feedback over the North Atlantic (Evan et al., 2016). Saharan dust significantly enhances nutrient transport to regions like the Amazon rainforest, which may also feedback on climate (e.g., Bristow et al., 2010; Yu et al., 2015). Over southern Africa, massive aerosol plumes during peak biomass burning are exported in a so-called "river of smoke" off the southeastern coast of southern Africa to the Indian Ocean, as well as over the southwestern coast over Angola out to the Atlantic Ocean (Garstang et al., 1996; Tyson et al., 1996a; Tyson et al., 1996b; Swap et al., 2003). This latter exit pathway aligns with the stratocumulus cloud deck that forms off of the southwestern coast and has motivated the NASA ORACLES aircraft campaign (https://espo.nasa.gov/oracles). The simulation of this cloud deck with the AeroCom intercomparison of global models was found to differ significantly between models, and to be the area of highest uncertainty in modeling aerosol radiative forcing (Stier et al., 2013). An assessment of the first AeroCom showed that the largest model diversities were from dust and carbonaceous aerosols (Kinne et al., 2006), the dominant aerosol constituents over Africa. Additionally, this AeroCom experiment highlighted an overestimation of dust at northern African sites in winter (Kinne et al., 2006). An accurate representation of African aerosols is critical in climate models to understand the regional and global radiative forcing and climate impacts of dust and biomass burning aerosols, at present and under future climate change, and is currently a major challenge.

This study performs the first evaluation of the representation of African aerosols in the Conformal Cubic Atmospheric Model (CCAM) (McGregor, 2005). The CCAM aerosol parameterizations are based on the CSIRO Mk3.6 climate model used in the CMIP5 intercomparison to estimate radiative forcing for the Intergovernmental Panel on Climate Change AR5, and CCAM will be included as part of a coupled earth system model, the Variable Resolution Earth System Model (VRESM), towards the South African Council of Scientific and Industrial Research (CSIR) submission to the sixth Coupled Model Intercomparison Project (CMIP6). We evaluate CCAM using the CMIP5 emissions inventory against long-term aerosol optical depth (AOD) observations across Africa and outflow regions off the coast from the Aerosol Robotic Network (AERONET) (Holben et al., 1998; Dubovik et al., 2002). A particular emphasis is placed on capturing the long-term seasonal variability at sites heavily impacted by dust and biomass burning aerosol particles. CCAM simulates four prognostic aerosol species (organic carbon (OC), black carbon (BC), sulfate, and dust) and non-prognostic sea salt aerosols, and their individual contributions to total AOD. Detailed case studies at six sites across Africa are used to examine the modeled source distribution of AOD and to understand the model processes, determining how well CCAM represents the observational data. The evaluation of aerosols in CCAM against observations has implications for its estimates of radiative forcing.





## 2 Methods

### 2.1 CCAM model description

CCAM is a global atmospheric model, and was used at a quasi-uniform global resolution of 50 km in the horizontal and with 27 levels in the vertical. The simulations applied in this study form part of the CSIR's contribution to the Coordinated

Regional Downscaling Experiment (CORDEX) of the World Climate Research Programme (WCRP). Horizontal wind and temperature upwards of 900 hPa and the surface pressure in CCAM were nudged towards the ERA-Interim reanalysis data (Dee et al., 2011). This nudging was applied every 6 hours at a length scale of ~2250 km using the digital filter of Thatcher and McGregor (2009). The sea-surface temperature and sea-ice from ERA-Interim were used; these values were interpolated to the CCAM grid with the differences in the land-sea mask taken into account. For this study, 6-hourly model output was

regridded to 0.5˚ x 0.5˚ resolution over the African continent (40˚N to 40˚S, 20˚W to 60˚E) from 1999 to 2012, when most AERONET observations are available for comparison. Prognostic soil variables like temperature and moisture, in addition to aerosol fields, were spun-up running the simulation for a year prior to the start of the experiment.

The aerosol parameterization in CCAM has been documented in detail elsewhere (Rotstayn et al., 2007; Rotstayn et al., 2010; Rotstayn et al., 2011; Rotstayn et al., 2012). In summary, the aerosol scheme is a bulk / mass scheme (i.e. single

moment) to represent the sulfur cycle, carbonaceous aerosols, dust, and diagnosed or non-prognostic sea-salt. The atmospheric model determines the transport of the prognostic aerosols, including turbulent mixing in the boundary layer and transport due to convection. Wet scavenging processes are included, with appropriate links to warm rain and frozen precipitation processes in the cloud microphysics parameterizations and the convection scheme (Rotstayn et al., 2007). The model also accounts for both direct and indirect aerosol effects, representing an important feedback into the atmospheric

simulation.

The model has three prognostic variables to represent the sulfur cycle: Dimethylsufide (DMS), $SO_2$ and sulfate. There are prescribed oxidant fields for OH, $NO_3$, $H_2O_2$ and $O_3$, to account for sulfur chemistry, with the amount of $SO_2$ dissolved into cloud water described by Henry's Law. Prognostic aerosol species for hydrophobic and hydrophilic forms of organic carbon (OC) and black carbon (BC) to represent carbonaceous aerosols. Hydrophobic OC and BC are non-hygroscopic,

while hydrophilic species' hygroscopic growth is based on Köhler theory. The size distribution of the sulfate, OC and BC aerosol particles is represented by a mode radius with a geometric standard deviation. Dust is represented by four size bins with radii of 0.1-1, 1-2, 2-3 and 3-6 µm, with the parameterization of aeolian dust emissions closely based on Ginoux et al. (2001) and Ginoux et al. (2004) (see also Rotstayn et al., 2011). Specifically, dust emissions are described by the expression,

$$F_p = CSs_p u_{10m}^2 (u_{10m} - u_T) \ \text{(if, u}_{10m} > \text{u}_t) \qquad (1)$$

where $F_p$ is the flux (µg s$^{-1}$ m$^{-2}$), $C$ is a dimensional factor set to 0.5 µg s$^2$ m$^{-5}$ , $Ss_p$ is a fraction for each dust size bin following Ginoux et al. (2001), $u_{10m}$ is the horizontal wind speed (m s$^{-1}$) and $u_t$ (m s$^{-1}$) is the threshold velocity, which accounts for soil moisture and the particle size. If $u_{10m}$ is not greater than $u_t$, then $F_p = 0$. For this study, the dimensional





factor $C$ was set to be smaller than that used by Ginoux et al. (2001), which has the effect of reducing the dust emissions for the same wind speed and soil moisture. The film droplet and jet droplet modes of sea-salt are diagnosed from 10m wind speed, rather than calculated using prognostic equations. Simulated sea-salt aerosol particles have two size bins.

Emissions of OC, BC and $SO_2$ from anthropogenic and biomass burning sources are from the CMIP5 recommended historical emissions datasets through the year 2000 (Lamarque et al., 2010) and extend through 2012 using emissions from the RCP8.5 low mitigation scenario (Moss et al., 2010; Riahi et al., 2007). Aerosol emissions across the RCP scenarios for the short time period studied here (i.e. 2005–2012) are similar (van Vuuren et al., 2011). Within CCAM, of the $SO_2$ emissions from fossil fuel and smelting, 3% are emitted as sulfate directly (Rotstayn and Lohmann, 2002); a similar fraction is assumed in other global models to represent rapid in-plume transformation of $SO_2$ to sulfate (Liu et al., 2005; Chin et al., 2000; Koch et al., 1999). Additional minor sources of model sulfate aerosol are volcanic $SO_2$ emissions and biogenic DMS emissions, which can be oxidized to sulfate (Rotstayn and Lohmann, 2002).

Within CMIP5, emissions from anthropogenic and biomass burning sources vary decadally, and during the 2005–2012 period forced by RCP8.5 vary every 5 years. Biomass burning emissions also have a monthly varying annual cycle, while anthropogenic emissions remain constant annually. Thus, changes in modeled aerosol loading using the CMIP5 emissions on temporal scales smaller than monthly for OC, BC and sulfate, and inter-annual variability within a given decade, are not due to changes in sources, but instead changes in transportation and deposition sinks resulting from meteorological variability. An earlier study over southern Africa during the biomass burning season found a chemical transport model was able to reproduce day-to-day variability in AOD using time-invariant emissions, suggesting meteorological variability is more important on this timescale than emissions (Myhre et al., 2003).

Hydrophobic and hydrophilic forms of BC and OC are transported separately in CCAM. The model assumes fossil fuel emissions are 50% hydrophilic, and biomass and biofuel burning are 100% hydrophilic. Conversion from hydrophobic to hydrophilic follows Cooke et al. (1999) with an e-folding lifetime of 1.15 days. Secondary organic aerosol (SOA) formation is not treated in the model. All prognostic aerosol species are removed via wet and dry deposition, while dust is additionally removed through gravitational settling (Rotstayn and Lohmann, 2002; Lohmann et al., 1999; Ginoux et al., 2001).

## 2.2 AERONET observational data

The global network of AERONET stations measure aerosol optical properties at a range of wavelengths (340 nm to 1020 nm) using a ground-based Cimel sun-photometer (Holben et al., 1998; Dubovik et al., 2002). For this work, the measured AOD at 440 nm ($AOD_{440nm}$), and the Ångström exponent of extinction for 440 nm to 870 nm ($\alpha_{ext(440/870)}$) from AERONET were used. AOD is the column-integrated attenuation of radiation due to aerosols from the surface to the top of the atmosphere (Eg. 2), where $\tau_\lambda$ is the AOD at wavelength $\lambda$, $z$ is the height (integrated from ground-level to the top of the atmosphere (TOA)), and $\sigma_{ext}$ is the aerosol extinction coefficient.

$$\tau_\lambda = \int_{z=0}^{z=TOA} \sigma_{ext}(\lambda, z)dz \qquad (2)$$





The Ångström exponent of extinction is the negative slope of the natural log of AOD with wavelength. The $AOD_{440nm}$ was adjusted to 550 nm using the $\alpha_{ext(440/870)}$ for comparison to modeled AOD following Eq. 3, where $\tau_{440}$ is AOD at 440nm measured by AERONET, and $\tau_{550}$ is AOD at 550nm:

$$\tau_{550} = \tau_{440} \left(\frac{550}{440}\right)^{-\alpha_{ext(440/870)}} \tag{3}$$

A climatology of $AOD_{550nm}$ and $\alpha_{ext(440/870)}$ observations from 34 sites Africa and the Middle East outside of heavily urbanized areas with at least 1 full year of level 2.0 data (cloud-screened, and manually inspected for quality assurance; Smirnov et al., 2000) (see Fig. 1 and Table 1) is developed. Sites that are heavily influenced by natural sources of aerosol particles were selected. In southern Africa sites to characterize model performance in regions dominated by biomass burning aerosol, and in northern and western Africa and the Middle East to characterize the model representation of Saharan and

Sahelian dust sources and outflow were prioritized. This includes sites in the Mediterranean and Europe influenced by North African dust outflow (Basart et al., 2009; Toledano et al., 2007a; Toledano et al., 2007b; Querol et al., 2009; Pace et al., 2006). This prioritization was because Africa has the largest emissions of dust in the world and the largest single source of biomass burning aerosols (Crutzen and Andreae, 1990; Schütz et al., 1981). Biomass burning is a major source of aerosol particles for the continent, contributing an estimated 86% of total BC and OC emitted in Africa, a higher percentage than

other regions worldwide (Bond et al., 2004).

For the comparison with model outputs, sites with multiple years of complete data for most of the annual cycle (see Sect. 2.3 and Fig. 3a and 3b) were selected. Where multiple sites were proximal to each other and showed similar features, the site with the longest data record was selected to be representative (see bolded site names in Fig. 1). This results in twenty-three sites used in the comparison with model outputs. Daily average values, calculated for days with at least 3

measurements, were downloaded from the AERONET website (http://aeronet.gsfc.nasa.gov) and used in this analysis.

## 2.3 Model-observation comparisons

Monthly-average time series and multi-year monthly mean climatology of $AOD_{550nm}$ were calculated for each site for observed and modeled data. The 550 nm wavelength is representative of the model AOD output. The AERONET monthly average $AOD_{550nm}$ was then calculated from the daily averages using a 70% data completeness rule where if more than 30%

of the daily values were missing, a monthly average could not be calculated for that time period. A multi-year mean seasonal cycle was also calculated from daily averages for each month for all available years of data at each site, following the same data coverage exclusions. This is to ensure the observed monthly averages were representative of the entire month to provide a relevant comparison for modeled output, as it is difficult for climate models to represent specific days individually (e.g., Magi et al., 2009), and as CCAM used CMIP5 emissions that do not vary daily.

For these comparisons, the temporal collocation of the observed and simulated AOD was considered, and were aligned as possible (Schutgens et al., 2016). In addition, the impact of the two datasets having different temporal sampling was assessed on the averaging periods of concern (e.g. monthly means). CCAM 6-hourly output was averaged for monthly and





multi-year means only between 06:00 and 18:00 UTC, as the AERONET sun-photometer measurements were only made during daytime (similar to other AERONET-model comparison studies; e.g., Tegen et al., 2013). Model monthly means were, however, insensitive to the choice of daylight cut-off (see Fig. 2), which gives confidence that the instantaneous 6-hourly values from CCAM can represent the full daytime hours sampled by AERONET, whose daily means are calculated

from a minimum of 3 time points during sunlight hours. Multi-year CCAM seasonal cycles were calculated at each site from 1) only the months with valid observational data, and 2) all model years (1999–2012). As many of the observational sites do not have continuous data, nor are the sampling times across sites overlapping or the same, the two calculations of modeled multi-year seasonal cycles were compared to test whether the entire model time period (1999–2012) for each month could be used to evaluate modeled spatial patterns against all available sites.

Modeled and observed $AOD_{550nm}$ at each site were compared on a monthly timescale using a variety of metrics to quantify how well the model captures seasonal and interannual variability, and overall magnitude. To this end, Pearson's correlation coefficient between the model and observations ($r$; the square root of the variance in the observations explained by the model), Normalized Mean Bias (NMB; Eq. 4) of the model as a percentage of the observed values, and the Mean Absolute Error (MAE; Eq 5.) of the model in units of $AOD_{550nm}$ were calculated. The NMB is calculated as follows, where $N$

is number of points, $M$ are modeled vales and $O$ are observed values:

$$NMB = \frac{1}{N}\sum_{i=1}^{N}\frac{M_i - O_i}{O_i} \times 100\% \qquad (4)$$

The MAE, where $N$ is number of points, $M$ are modeled vales and $O$ are observed values, is:

$$MAE = \frac{1}{N}\sum_{i=1}^{N}|M_i - O_i| \qquad (5)$$

**3 Climatology of AERONET AOD over Africa: seasonal variability and its drivers**

Figures 3a and 3b show a compilation of multi-year monthly mean observed $AOD_{550nm}$ and Fig. 4a and 4b $\alpha_{ext(440-870)}$ values for the 34 study sites, ordered by region from north to south. The symbols are the multi-year mean values, and the whiskers represent ±1 standard deviation. The number of years of data used per month is shown at the top of the plot area. The Ångström exponent is an empirical representation that can give information on particle size, with values varying

between approximately 0 for coarse dust particles to 2 for submicron particles (Leon et al., 2009; Hamonou et al., 1999). The Ångström exponent values presented here are based on aerosol extinction. In Fig. 4a and 4b, values of $\alpha_{ext(440-870)}$ below 0.4 are indicative of aerosols dominated by coarse mineral dust (shaded gray area), while higher values show a contribution from fine, submicron aerosols, indicative of biomass burning or anthropogenic sources (Holben et al., 2001; Ogunjobi et al., 2008; Rajot et al., 2008).

Table 1 displays the multi-year daily average $AOD_{550nm}$ and $\alpha_{ext(440-870)}$, which were calculated using all available data points per site. In addition, the maximum and minimum multi-year monthly average value per site is displayed together with





the month when that value was measured. The amount of data is not equal at all sites, nor were the sampling periods at all sites overlapping, and thus detailed comparisons of the sites are not possible. Instead, the focus will be on overall regional trends, including timing of peaks and minima.

### 3.1 Northern Africa and Middle East

5     The mean $AOD_{550nm}$ values in the northern African sites (blue in Fig. 1 and Fig. 3a) range from 0.06–0.49, with maximum values ranging 0.15–0.69, and minimum values ranging 0.015–0.36. The $\alpha_{ext(440-870)}$ values average 0.49–1.04, with maximum values ranging 0.73–1.59 and minimum values ranging 0.17–0.96. The spread of $\alpha_{ext(440-870)}$ values suggests a mixture of fine and coarse aerosols at these sites.

     The impact of dust on the aerosol loading is observed at these sites. Ras El Ain, Ouarzazate, La Laguna, Dahkla, Solar 10  Village, Mezaira, Hamim, and Tamanrasset INM have multi-year monthly average $\alpha_{ext(440-870)}$ below the 0.4 "dust" threshold, and all other sites pass this threshold within the standard deviation from the multi-year mean except for El Arenosillo. This may be due to the influence of local industrial pollution sources there (Toledano et al., 2007a; Toledano et al., 2009).The maximum $AOD_{550nm}$ occurs across most sites during June–August, and coincides with a decrease in $\alpha_{ext(440-870)}$. This is later than the $AOD_{550nm}$ peak at the western African sites (Sect. 3.2). This delay and corresponding change in $\alpha_{ext(440-870)}$ suggest 15  that transported dust from the Sahara leads to the higher observed $AOD_{550nm}$. Thus, the seasonal variation in the location of the ITCZ and associated northward shift in dust transport may be responsible for the shift in timing of maximum $AOD_{550nm}$ between the western and northern African sites. $AOD_{550nm}$ at most of the Middle Eastern sites (Eilat, Sede Boker, IASBS, KAUST, and Solar Village) peaks earlier, in March through May, indicative of different seasonality of the local dust sources in the Arabian peninsula (Basart et al., 2009).

20     The greatest seasonal differences in $\alpha_{ext(440-870)}$ occur at Hamim, where in addition to high local dust emissions in spring and summer, regional circulation transports dust from deserts in Iraq and Southern Iran during summer and a mixture of fine pollution aerosols from the Persian Gulf throughout the year (Eck et al., 2008; Basart et al., 2009). The Izana site has a different seasonal pattern in $\alpha_{ext(440-870)}$ than its neighboring two sites, La Laguna and Santa Cruz, on the same island. It is, however, the highest elevation site in our study at 2391 m, 1800–2300 m higher than La Laguna and Santa Cruz (see Table 25  1). Local topography, meteorology, or transport patterns affecting the sinks and sources reaching Izana may lead to a different aerosol size distribution.

### 3.2 Western Africa

     The highest $AOD_{550nm}$ across all sites is observed in western Africa (denoted in red in Fig. 1 and Fig. 3b). The overall mean $AOD_{550nm}$ ranges 0.44–0.67. $AOD_{550nm}$ values peak at 0.62–1.10, and minimum $AOD_{550nm}$ values range 0.26–0.38. The 30  minimum $AOD_{550nm}$ values seen here are similar to the maximum $AOD_{550nm}$ values seen in northern and southern Africa. The western African sites also have low $\alpha_{ext(440-870)}$ values across most months (0.29–0.66) with maximum values ranging




0.52–0.96, and minimum values ranging 0.092–0.33. The maximum multi-year monthly average $\alpha_{ext(440-870)}$ value occurs in December across all western Africa sites, while the minimum values vary in timing.

In general, as $AOD_{550nm}$ values increase, $\alpha_{ext(440-870)}$ values decrease to low values, which would suggest that the variation in the $AOD_{550nm}$ is dominated by the variation in coarse dust aerosol. A similar relationship was found previously for Banizoumbou (Holben et al., 2001; Ogunjobi et al., 2008; Rajot et al., 2008). This relationship is prominent at Agoufou, Banizoumbou, Zinder Airport, Maine Soroa, and Ougadougou. In addition, this relationship is seen in January–June in Djougou, while in October–December the increase in $AOD_{550nm}$ at this site corresponds to an increase in $\alpha_{ext(440-870)}$. In Ilorin, which is south of the other sites, the $AOD_{550nm}$ peaks in January–March, while the $\alpha_{ext(440-870)}$ is at a minimum value in March–May.

The timing of peak monthly-mean $AOD_{550nm}$ varies between February–March for the Banizoumbou, Ouagadougou, Djougou, and Ilorin sites, and May–June for the Agoufou, Dakar, Zinder Airport, and DMN Maine Soroa sites, approximately following a south to north gradient. The latitudinal movement of dust transport northward from winter (i.e. February–March) to summer (i.e. May–June), thus appears to dictate the seasonal cycle in AOD at these sites, consistent with a previous regional dust model-AERONET comparison at Dakar, Agoufou, and Banizoumbou (Tegen et al., 2013).

Ilorin and Djougou, the most southerly sites in this region, have slightly higher $\alpha_{440-870}$ values on average (0.66 ± 0.36 and 0.52 ± 0.34, respectively), especially during late fall to early winter (peaking at ~0.9 in December). This coincides with the sub-Sahelian Northern Africa biomass burning season (December–February) (e.g., Roberts et al., 2009; Giglio et al., 2006). The highest $AOD_{550nm}$ during December–February out of the western African sites is also observed at Ilorin and Djougou (up to a peak of 1.10 in February at Ilorin), which are closer to the primary area of biomass burning during this time (Liousse et al., 2010; Pinker et al., 2010). This suggests that biomass-burning aerosols could make up a larger fraction of total $AOD_{550nm}$ at Ilorin and Djougou than elsewhere during this time period.

Dakar has the smallest month-to-month variability in $AOD_{550nm}$, ranging from 0.30–0.62. Leon et al. (2009) find that Dakar is subject to transport of both dust and biomass burning aerosols, depending on the season, as well as poorly constrained anthropogenic emissions from the city and other nearby urban centers; thus Dakar is influenced by a variety of sources. The site's greater distance from the natural aerosol sources and proximity to anthropogenic emissions that have lower seasonal variability may explain its observed seasonal cycle.

### 3.3 Southern Africa

The average $AOD_{550nm}$ in the southern African sites range 0.064–0.21, with maximum $AOD_{550nm}$ peaking at 0.095–0.50 and minimum $AOD_{550nm}$ ranging 0.046–0.13. The region has larger $\alpha_{ext(440-870)}$ values, with averages ranging 0.7–1.6. The maximum monthly averages range 1.12–1.85, and the minima range 0.28–1.14. Mongu and Skukuza in southern Africa have the highest observed $\alpha_{ext(440-870)}$ values, indicating little influence from coarse dust and confirming the importance of biomass burning as an aerosol source in this region.




Previous studies have shown AOD is highest in this region during the biomass burning season, from AERONET AOD through the year 2007 at Mongu and Skukuza (Queface et al., 2011) and MISR satellite data over South Africa (Tesfaye et al., 2011). Mongu is situated in Zambia in the middle of the biomass burning source region in southern Africa (e.g., Swap et al., 2003; Eck et al., 2003; Edwards et al., 2006; Queface et al., 2011). For Southern Hemisphere Africa, peak fire activity

typically occurs in June through October, with a shift in general toward later months moving from north to south, except in the winter rain areas of southwestern South Africa (Archibald et al., 2010; Giglio et al., 2006).

At Ascension Island, the transport of biomass burning aerosols from southern Africa west over the Atlantic Ocean is observed in the seasonal cycle of $\alpha_{ext(440-870)}$ and $AOD_{550nm}$, as both peak in September, which is the timing of climatological peak AOD and peak biomass burning at Mongu (Giglio et al., 2006). This known transport pathway off the coast of Angola

(Garstang et al., 1996) is also seen in the $AOD_{550nm}$ and $\alpha_{ext(440-870)}$ observed at Etosha Pan, but peak values occur in October as opposed to September. However, these values at Etosha Pan may not represent a long-term mean seasonal cycle as only one year of data was available at this site.

The $AOD_{550nm}$ at Skukuza also peaks in September, indicating transport of biomass burning aerosols southeast over the site and exiting the continent toward the Indian Ocean, consistent with the so-called "river of smoke" or major export

pathway off the coast of southeastern South Africa (e.g., Swap et al., 2003). Although Reunion Island is not within this path, evidence of eventual transport of biomass burning aerosols from southern Africa is apparent in the seasonal cycle of $\alpha_{ext(440-870)}$ and $AOD_{550nm}$, which increase toward an October peak.

The continental sites closest to the region of burning have sustained and relatively constant high values of $\alpha_{ext(440-870)}$ during April–October. This is especially evident at Mongu. The $\alpha_{ext(440-870)}$ at all southern Africa sites declines in austral

spring and summer. While these small variations in $\alpha_{ext(440-870)}$ alone are not enough to distinguish aerosol size distributions, they are consistent with results from MISR for the central South African region (including Skukuza) that showed an increase in the coarse mode fraction in summer due to dust from the Northern Cape and Namibian desert regions (Tesfaye et al., 2011).

## 4. Model evaluation

### 4.1 Annual model aerosol budgets

Annual burdens, deposition, wet deposition fraction, lifetime, and emissions for each of the four prognostic aerosol species in 2010 are shown in Table 2 for the globe and the Africa domain (40˚S to 40˚N, 20˚W to 60˚E), separately. These values are compared to estimates from other present-day models and the CMIP5 and AeroCom experiments in Fig. 5.

CCAM is within the range of global present-day annual aerosol burden estimates from models within CMIP5 and

AeroCom experiments for BC, OC, and sulfate. In addition, in Fig. 5b–c, CCAM is within the range of estimates for total deposition, fraction wet deposition, burden and lifetime of organic aerosols (OA) and BC (Tsigaridis et al., 2014; Allen and Landuyt, 2014). CCAM modeled OC emissions and burden is converted to OA by multiplying by a factor of 1.4 for a





consistent comparison (Tsigaridis et al., 2014). In general the CCAM values for BC are higher than the CMIP5 median values, but are well within the range of models. For OA, CCAM is close to median estimates from the AeroCom Phase II models with the exception of OA/OC lifetime, which is at the high end of all models.

While CCAM performs well compared to other models for BC, OC and sulfate, CCAM has a dust burden (68 Tg) ~2–7

times higher than AeroCom Phase I models (Huneeus et al., 2011) and all available dust modeling results summarized in a recent review (Kinne et al., 2006; Zender et al., 2004) (see Fig. 5a, 5d). In the CCAM model, annual dust emissions over the Africa region alone (40˚S to 40˚N, 20˚E to 60˚W) in 2010 are 2320 Tg $yr^{-1}$, contributing 83% of global total modeled dust emissions. The range from AeroCom models is 35–77.9% of global dust emissions (Huneeus et al., 2011). Global dust emissions (Fig. 5) are above the mean but within the range of AeroCom models. This together with an overestimation of dust

in Africa would lead to a large percentage contribution of global dust emissions from Africa.

The global dust emissions, burden, wet deposition, dry deposition and sedimentation, and lifetime are compared to AeroCom experiments in Fig. 5d (Huneeus et al., 2011). The modeled dust lifetime (8.9 days) is longer than models examined in Zender et al. (2004) that range from 2.8 to 7.1 days, and AeroCom Phase I that range from 1.6 to 7.1 days (Huneeus et al., 2011), indicating the sinks of dust in the model may be too low, contributing to a high global dust burden.

The wet deposition (1571 Tg $a^{-1}$) is higher than AeroCom results (range of 295 to 1382 Tg $a^{-1}$, median 357 Tg $a^{-1}$), however the dry deposition and sedimentation (1209 Tg $a^{-1}$) are similar to the AeroCom median (753 Tg $a^{-1}$) in spite of the much higher dust burden. This overestimation of dust is discussed more in Sect. 4.2.2 and Sect. 4.2.3 below.

## 4.2 Evaluation of model against observations: Multi-year mean seasonal cycle comparison

Figure 6 shows the same observed multi-year mean seasonal cycle as in Fig. 3 (here in red triangles), overlaid with CCAM results for all model years (dark blue) and only those months with corresponding AERONET data that met the 70% completeness cutoff (yellow). The shaded red areas are ±1 standard deviation from the observed values, and the shaded blue areas are ±1 standard deviation from the all model years CCAM output. In this comparison, only AERONET sites with multiple years of complete data for most of the annual cycle are included in order to compare multi-year monthly cycles

from observations and the model.

The monthly cycle from CCAM considering the full model period (dark blue line) and only those years with observational data (yellow line) are similar across all sites, with only minor differences that are within ±1 standard deviation of the full model period. Thus, the full model time period (1999–2012) can be used to evaluate modeled spatial patterns against all available AERONET sites, even though the observations at different sites are from disparate time periods. All

following analyses are presented using the full model time period.

For most sites, the monthly cycle (i.e. timing of peak and minimum $AOD_{550nm}$ values) is well-captured by CCAM, indicating the seasonality in CMIP5 emissions and the model parameterization of dust emissions is adequate. A few notable exceptions (e.g., timing of maxima at Mongu and Ascension Island, missing winter minima in western African sites, and



spurious summertime peaks after observed springtime maxima at Sede Boker and Solar Village) will be investigated in Sect. 4.2.1–4.2.3 below. The magnitude of modeled and observed multi-year monthly average $AOD_{550nm}$ overlap within ± 1 standard deviation of each other for at least 7 months at all sites except Reunion Island, and for all observed months at 8 sites that span all three regions (Granada, Blida, Zinder Airport, Banizoumbou, Ouagadougou, Djougou, Ilorin, and Skukuza). 5   The differences in magnitude per region will also be detailed in Sect. 4.2.1–4.2.3 below.

Figure 7 highlights two representative sites each from the northern, western, and southern regions with the most observational data available in greater detail, comparing multi-year monthly mean observed and modeled $AOD_{550nm}$, with the modeled contribution of each aerosol type (sea salt, large size bin dust (radius ≥ 1 μm),, small size bin dust (radius < 1 μm), BC, OC, sulfate) to total $AOD_{550nm}$ shown.) Further investigation of model performance, by region, follows.

10   **4.2.1 Southern Africa**

The model generally represents the magnitude of $AOD_{550nm}$ at all southern African sites outside of Reunion Island (see also Fig. 6, Table 3). However, the timing of the modeled peak $AOD_{550nm}$ at two of the sites where maximum $AOD_{550nm}$ is dominated by biomass-burning (Ascension Island and Mongu) occurs 1 month too early (in August, instead of September as highlighted in Table 1). Modeled $AOD_{550nm}$ at both Mongu and Skukuza remain relatively constant between August and 15   September (Fig. 7). This is consistent with the observations at Skukuza, likely due to the greater influence of anthropogenic aerosol sources at this site. Figure 7, shows the modeled sulfate contribution (emitted from both anthropogenic and biomass burning) to total $AOD_{550nm}$ is higher and that of OC (primarily emitted from biomass burning) is lower at Skukuza relative to Mongu, indicating the breakdown of model emissions sources is consistent with this explanation. There is a larger observed increase in $AOD_{550nm}$ between August and September at the biomass burning source region (Mongu) and the more remote 20   Ascension Island whose seasonality is impacted by transported biomass burning aerosol as seen in the $\alpha_{ext(440-870)}$ (Fig. 4a and 4b).

This mismatch in timing of the peaks is a long-standing issue in understanding southern African biomass burning, first noted during the SAFARI-2000 measurement campaign (Swap et al., 2003). In a study of Southern Hemisphere biomass burning observed by satellite, Edwards et al. (2006) found that in southern Africa alone, peak CO and AOD lagged peak fire 25   counts by ~1 month (late September to October vs. late August, respectively). Using a chemical transport model, they found that the residence time of CO over the region was much too short for transport patterns to explain the 1 month time lag (Edwards et al., 2006). Two recent modeling studies also found that peak AOD over Southern Hemisphere Africa lagged peak fire counts and estimates of peak biomass burning emissions using either the GFEDv2 or AMMA inventories by 1–2 months (Magi et al., 2009; Tummon et al., 2010). The CMIP5 emissions used in our CCAM model study are from GFEDv2 30   for year 2000 onward (van der Werf et al., 2006; Lamarque et al., 2010), which at the source region of Mongu peak in August leading to the maximum modeled AOD. The GFED inventory is based on estimates of burned area from burn scars and thermal signatures of active fires viewed by the MODIS satellite, combined with land cover data and meteorological parameters to estimate emissions for different vegetation types (van der Werf et al., 2006; van der Werf et al., 2010). This





type of method would only capture large fires that produce satellite-detectable burn scars. A recent study updated the GFED inventory to include a parameterization of fire counts, burned area, and emissions from previously missing small fires, but this did not change the seasonality in biomass burning emissions over Southern Hemisphere Africa (Randerson et al., 2012). Burned area still peaked in August, as it increased more early in the biomass burning season than late in the season when

small fires were included, and higher fuel load burns (e.g., from dense, wooded vegetation) late in the season did not lead to a compensating change in emissions (Randerson et al., 2012). The small fires parameterization still relies on detection of thermal anomalies (Randerson et al., 2012).

The $AOD_{550nm}$ peak in September aligns with the peak in fire intensity found in the generalized fire regime of savanna-woodland in Archibald et al. (2010). The peak in fire intensity in southern Africa as well as fire size occurs later in the

season than the peak in fire number, though the increase in these is not large over the season (Archibald et al., 2010). However, this does suggest that fire intensity may be an important factor to consider in modeling emissions from biomass burning in southern Africa, e.g., through the new initiative FireMIP (Hantson et al., 2016).

Table 3 displays a summary of model-observation comparison by site. The normalized mean bias of the model is negative at Mongu (-21.2%) and positive at the three other southern Africa sites, showing that overall $AOD_{550nm}$ is

underestimated at the source while overestimated at receptor regions (Table 3). Figure 6 suggests the model overestimates transport of biomass burning emissions to receptor sites in particular for the months of June through August. Because the $AOD_{550nm}$ values in both the model and observations are smaller here than in other regions, the mean absolute error is very low (0.07–0.09) and is the lowest of all sites in this model comparison. At all sites except Reunion Island, the model captures some of the temporal variability, with highly statistically significant correlation coefficients ranging from 0.48 to 0.67.

Relative to other regions, the model performs best over southern Africa in terms of mean $AOD_{550nm}$ magnitude, but overestimates the transport of biomass burning aerosols to Reunion Island in June through September.

### 4.2.2 Western Africa

At the western African sites, which in the observations are dominated by dust (Fig. 4b), the model captures the overall seasonal cycle in $AOD_{550nm}$ except between September and December, where the observations show a decrease at all sites

except the two southernmost (Djougou and Ilorin) while the model increases (see Fig. 6). As a result, the modeled minimum $AOD_{550nm}$ occurs between August and October, instead of in November–December as in the observations at Agoufou, Dakar, Zinder Airport, Banizoumbou, DMN Maine Soroa, and Ouagadougou.

Figure 7 shows in a case study for two sites, Dakar and Banizoumbou, the strong influence of dust on these sites. The increase in modeled $AOD_{550nm}$ from September through December, which is not seen in the observations, is due to increases

in the large dust (orange bars) and small dust (red bars) contribution. This could be due to the systematic overestimate of 10m wind speed during the dry season in several meteorological re-analyses in the Sahelian region (Largeron et al., 2015). Although the ERA-Interim reanalysis used in this study was found to perform best overall against wind speed observations,





it also exhibited a strong positive bias during Northern Hemisphere winter (Largeron et al., 2015), which would lead to an overestimate in wind-driven dust emissions during this season (September–December).

The remainder of the shape of the seasonal cycle is captured relatively well at western African sites, with the peaks in $AOD_{550nm}$ in CCAM occurring within 1 month of the peak in AERONET $AOD_{550nm}$. Only at Ilorin is the timing of the peak

the same in the model and the observations. The Pearson's correlation coefficient between the modeled and observation $AOD_{550nm}$ are statistically significant ($r$ ranges 0.27–0.61) at all sites except Djougou (Table 3). The lack of statistically significant correlation at Djougou may in part be due to a lack of data with only 24 individual months. In most of the western African sites, the model has an overall positive normalized mean bias (ranging from 29% to 103%). The exceptions are Djougou and Ilorin, which are the two southernmost sites. Djougou and Ilorin are slightly farther away from major dust

sources originating in topographic depressions (Evan et al., 2015), which are represented in the CCAM dust emissions scheme (Rotstayn et al., 2011), and have relatively small, but negative normalized mean biases (-1.3%, -12.6%, respectively). The mean absolute error for all sites ranges from 0.20–0.48, which are higher than southern Africa, but lower than northern Africa, which has lower $AOD_{550nm}$ on average compared to the western African sites.

The model overestimates in $AOD_{550nm}$ at western African sites closer to the dust source regions may be due to an

overestimate of wind speeds. Largeron et al. (2015) found that on an annual mean scale, ERA-Interim overestimates observed 10m wind speeds by 0.27 m s$^{-1}$ß in the Sahel, but this was largely a result of the wintertime overestimate mentioned previously. In fact, wind speeds during springtime and the monsoon season were underestimated in the ERA-Interim because the reanalysis did not represent large increases in wind speed from boundary layer free convection and deep convection (Largeron et al., 2015). Previously, the CSIRO Mk3 coupled GCM accounted for this by estimating sub-grid gustiness from

both boundary layer and deep convection to increase the effective 10m wind speed used in the model dust emission parameterization (Ginoux et al., 2004). In the case of CCAM, it was found that the effective sub-grid scale winds were too high, possibly due to differences in vertical and horizontal resolution, as well as changes in the model physical parameterizations. This led to an overestimation of global total dust emissions that were far outside the range suggested by observations (Rotstayn et al., 2011; Rotstayn et al., 2012). Therefore, these sub-grid gustiness terms have been removed

from the model version presented here. In spite of this, it is still possible that 10m winds in the model may be inaccurate, as the horizontal wind fields are only nudged to ERA-Interim above 900 hPa, not down to the surface, and at a coarse scale (2250 km resolution; see Sect. 2.1). Part of the determination of surface wind speeds in CCAM relies on the Community Atmosphere-Biosphere Land Exchange (CABLE) model estimate of surface roughness. Dust emissions additionally depend on local soil moisture and soil texture from the CABLE land surface model. Issues with modeled precipitation, the response

of soil moisture to precipitation, and how recent changes to soil texture implemented in CABLE from the Harmonized World Soil Database affect the atmospheric simulation could contribute to an overestimate in dust emissions.




### 4.2.3 Northern Africa and Middle East

Potential issues with dust emissions and transport in CCAM become more apparent when comparing to northern African $AOD_{550nm}$ observations. There are substantial overestimates of the multi-year monthly mean $AOD_{550nm}$ values in northern Africa (see Fig. 6) of up to a factor of 8 to 42 for individual months at each site. This region has the highest normalized mean

biases, with NMB over 200% at 6 of the 11 sites (see Table 3). As shown in Fig. 7 for two of the northern sites, Saada and Izana, almost all modeled $AOD_{550nm}$ in this region comes from dust. However, the observational data indicate that Saada and Izana rarely experience low values of $\alpha_{ext(440/870)}$ reaching the threshold representative of coarse dust (Fig. 4a). Thus, CCAM overestimates the contribution of dust to $AOD_{550nm}$ over Saada and Izana. The global dust burden in CCAM (67 Tg) is more than twice that of the high end of values in a recent review of global dust models as well as AeroCom and CMIP5 models

(Zender et al., 2004). Global dust emissions are higher than the median but are well within the range of estimates from Zender et al. (2004) and AeroCom models (Huneeus et al., 2011) (see Fig. 5). It is possible that an overestimate of dust lifetime combined with an overestimate of dust emissions plays a major role in this issue (see Section 4.1). At the same time, over the Arabian Peninsula (Dhadnah, Solar Village, Hamim) the model performs better with the lowest mean biases across sites in northern Africa and the Middle East (Table 3), suggesting dust emissions and transport may be better characterized in

this region.

However, the model does capture the monthly trends in observed $AOD_{550nm}$, with a strong peak in boreal summer and relatively lower values through rest of the year. At Saada, Izana and Dahkla, CCAM $AOD_{550nm}$ peaks in August, while the observations peak in July. Modeled and observed $AOD_{550nm}$ peaks in June at Hamim and July at Blida and Dhadnah. At Tamanrasset INM, CCAM $AOD_{550nm}$ also peaks in July, however there are no data for July at that site. The model output

shows a higher proportion of dust $AOD_{550nm}$ relative to total $AOD_{550nm}$ in the summer months, especially July and August (Fig. 7), which is consistent with the observed decrease in $\alpha_{ext(440/870)}$ and known northward movement of Saharan dust transport in summer from the shifting ITCZ (Jankowiak and Tanre, 1992; Moulin et al., 1997; Leon et al., 2009; Schepanski et al., 2009). The model also reproduces the increase in fine aerosol (e.g., BC and $SO_4$) relative to coarse dust in winter months at the two sites (Fig. 7) as implied by the increasing observed $\alpha_{ext(440/870)}$ (Fig. 4a). In spite of the high model bias, all

sites in northern Africa and the Middle East have statistically significant correlations, including some of the highest correlation coefficient values (ranging from 0.23 to 0.89). At Sede Boker, which has the lowest correlation coefficient in this region, the model predicts an increase in $AOD_{550nm}$ from June to August, similar to other Northern African sites, which is not observed. This discrepancy may be caused by an overestimate of Saharan dust transported to the site during summer.

### 4.2.4 Spatial patterns

Figure 8 shows the multi-year monthly mean climatology of modeled (background) and observed (filled circles) $AOD_{550nm}$ for March (Fig. 8a), representing high values of $AOD_{550nm}$ at many western African sites, and September (Fig. 8b), the peak observed $AOD_{550nm}$ at many southern Africa sites impacted by biomass burning (note the different scales for the




two months). Panels showing the 5$^{th}$ and 95$^{th}$ percentiles of 6-hourly CCAM AOD$_{550nm}$ highlight the modeled variability and additional spatial features. The scales are consistent across the maps within each month to aid in comparison and as such some high AOD$_{550nm}$ values are saturated in the color scale (see legend in Fig. 8).

We take advantage of the high temporal and spatial resolution of the model to show how significantly an individual 6-hourly output, in this example within the months of March and September (Fig. 8), can depart from the multiyear monthly mean AOD$_{550nm}$. Given that the emissions of all aerosol species and their precursors (with the exception of dust) vary only on a monthly to multi-annual timescale in CMIP5 emissions (see Sect. 2.1) the variability at the 6-hourly timescale must be a result of transport and aerosol sinks in the model (and dust emissions for western and Northern Africa). This confirms the importance of model processes driven by meteorology to modeled AOD$_{550nm}$. In Southern Hemisphere Africa, where

aerosols are dominated by emissions that in the CMIP5 emissions inventory are constant within a given month for a 5 to 10 year period, Fig. 8b shows that fine-scale temporal variability can still be represented in spite of limitations in emissions inventories, consistent with previous work in this region (Myhre et al., 2003).

    In March (Fig. 8a), the discrepancy in the modeled location of maximum AOD$_{550nm}$ in dust-dominated northern and western Africa and the Middle East is clear, as CCAM overestimates mean AOD$_{550nm}$ at all sites in this region except

Djougou and Ilorin, the two southernmost sites, which are underestimated. Given that the large-scale circulation in the model is constrained to reanalysis data, it seems unlikely that issues with large-scale transport would lead to this spatial pattern in the misrepresentation of AOD$_{550nm}$. It is more likely that the overall overestimate in total dust emissions varies regionally due to regional discrepancies in precipitation, soil texture, and soil moisture, that contribute to the surface roughness (which affects surface wind speeds, feeding into the magnitude of dust emissions) and dryness (which determines the likelihood of

erosion and dust emission). Dust emissions may be especially overestimated towards the north and northwest of Africa, and may even be locally underestimated in the southern Sahel. Figure 8a also shows modeled AOD$_{550nm}$ over the Arabian Peninsula is more consistent with observations, suggesting a better model representation of local dust emissions in this region (see also Sect. 4.2.3). In September, when AOD$_{550nm}$ is less impacted by dust, CCAM better captures the mean AOD$_{550nm}$ at the available sites in western Africa along a similar latitude band, but still significantly overestimates AOD$_{550nm}$

at more northern sites. This also points to a regional overestimate in dust emissions. Modeled dust lifetime may also play a role, which is longer than CMIP5 and AeroCom models (see Fig. 5 and Sect. 4.1) and could lead to dust transported too far off the northern coast of Africa. Another climate modeling study found that a non-prognostic dust scheme resulted in dust shifted too far north, while prognostic dust simulations had too much dust transport off the coast of western Africa (Mulcahy et al., 2014), suggesting the interaction between dust and meteorology in the model may be important in the current study as

well. This is an area of on-going study in CCAM.

    Modeled AOD$_{550nm}$ at the biomass burning source region, Mongu, is slightly underestimated in September, as seen in Fig. 8b, but the mean modeled AOD$_{550nm}$ values at receptor regions like Ascension Island and Skukuza are similar to the observed values. The transport in CCAM of biomass burning aerosols off the coast of Angola and southeastern Africa is visible with small enhancements in the mean modeled AOD$_{550nm}$, but is more apparent in the 5$^{th}$ and 95$^{th}$ percentile results,



showing that the model captures known exit pathways for southern African biomass burning (Garstang et al., 1996; Swap et al., 2003). Figure 8b also illustrates that the transport of biomass burning aerosols from southern Africa eastward toward Reunion Island is overestimated. Overall, from this analysis and given that the lifetime of OC and BC aerosols in CCAM is more consistent with other global models from AeroCom and CMIP5 (Tsigaridis et al., 2014; Allen and Landuyt, 2014),

biomass burning aerosol emissions and transport are relatively well represented in CCAM driven by the CMIP5 emissions inventory.

## 5 Conclusions

The compilation of long-term AERONET observations across Africa indicates different regimes of source types and their seasonality for northern, western, and southern Africa. The importance of dust and biomass burning aerosols in the

regions, as well as the transport and long-range impact of these aerosol sources, are evident in the $AOD_{550nm}$ and $\alpha_{ext(440-870)}$ trends across sites.

The prognostic aerosol scheme in CCAM is a key feature in the coupled earth system model VRESM currently under development for inclusion in the CMIP6 intercomparison. An accurate representation of African aerosols is critical in climate models and this current evaluation to understand how well the scheme performs in the present-day when forced with

CMIP-style emissions is essential to interpreting any future climate predictions using the model. CCAM aerosol output for OC, BC and sulfate compares well with output other CMIP5 models and AeroCom model global experiments. CCAM captures the seasonal cycle of the $AOD_{550nm}$ well at most sites, with statistically significant correlation coefficients between the model and observed monthly mean timeseries of $AOD_{550nm}$ at all but two sites of the 23 sites studied. The seasonal cycle at these sites is strongly influenced by dust and biomass burning aerosols, and thus CCAM is able to capture the general

seasonal cycle of the emissions of dust, and the transport of all aerosol types.

This analysis has also highlighted areas within CCAM and the emissions inventory that need further work. There is a notable shift in peak $AOD_{550nm}$ one month earlier than observations in biomass burning regions. This shift has been seen in previous modeling studies, and is likely due to missing processes in the emissions inventory. Comparing to CMIP5 models and AeroCom global experiments, CCAM overestimates many dust parameters including burden and lifetime. This

overestimate is also seen in the comparisons to AERONET at northern and western African sites. At the northern African sites in particular, the model has large positive normalized mean biases. The model attributes large $AOD_{550nm}$ values primarily to dust where the observations of the Ångström exponent and $AOD_{550nm}$ suggest there is very little dust present. This is likely a combination of an overestimate of dust lifetime leading to longer-range transport of dust and higher dust burdens, and overestimated dust emissions in the northwestern Sahara. The increase in $AOD_{550nm}$ in the boreal winter at

western African dust-influenced sites is likely due to a high bias in ERA-Interim reanalysis wind speeds in the Sahel during this season (also present in other reanalyses). The simulation of local soil parameters and injection height in CCAM could



also lead to emissions biases; testing and improvement of these fields in the CABLE land surface model in the development of VRESM may help to improve the representation of dust aerosols in Africa.

The CCAM results are consistent with state-of-the-art CMIP5 GCMs, providing confidence for using the model to study the regional impacts and linkages between African aerosols and climate change under different scenarios. In addition, 5 CCAM can be used to downscale the CMIP5 GCMs to finer spatial scales with its variable resolution global grid, and therefore refine our understanding of aerosols in this important region.

**Author contribution**

10 H. Horowitz, R. Garland, M. Thatcher and F. Engelbrecht developed the research question and designed the experiment. H. Horowitz performed the analyses of AERONET and CCAM data. M. Thatcher developed the prognostic aerosol scheme in CCAM. W. Landman provided input into the model-observation comparisons and needed statistical test. J. van der Merwe and Z. Dedekind extracted and formatted CCAM data. H. Horowitz prepared the manuscript with input from all authors.

**Competing interests**

15 The authors declare they have no conflict of interest.

**Acknowledgements**

This work was supported by NRF CSUR Grant Number 9157, and a CSIR Parliamentary Grant. HH was funded through the NSF GROW with USAID RI Fellowship. We thank the PIs and their staff for establishing and maintaining the 34 AERONET sites used in this study.



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





**Table 1: AERONET site information; bolded sites are used in model comparison. Average, maximum and minimum values for AOD$_{550nm}$ and $\alpha_{ext(440/870)}$ per site.**

| Site | Lat (°N) | Lon (°E) | Elevation (m) | Years of data used in study | AOD$_{550nm}$ [a] Multi-year daily average | Max multi-year monthly average ± 1 sd (month) | Min multi-year monthly average | $\alpha_{440\text{-}870}$ [a] Multi-year daily average | Max multi-year monthly average | Min multi-year monthly average |
|---|---|---|---|---|---|---|---|---|---|---|
| **Granada** | 37.16 | -3.61 | 680 | 2005–2012 | 0.15 ± 0.11 | 0.19 ± 0.13 (Aug) | 0.083 ± 0.041 (Jan) | 1.04 ± 0.46 | 1.59 ± 0.26 (Jan) | 0.67 ± 0.36 (Aug) |
| El Arenosillo | 37.11 | -6.73 | 0 | 2000–2009 | 0.14 ± 0.11 | 0.17 ± 0.13 (Sep) | 0.088 ± 0.052 (Dec) | 1.04 ± 0.42 | 1.38 ± 0.43 (Jan) | 0.96 ± 0.41 (Apr) |
| SAGRES | 37.05 | -8.87 | 26 | 2011–2012 | 0.12 ± 0.10 | 0.17 ± 0.23 (Jun) | 0.080 ± 0.036 (Jan) | 0.82 ± 0.35 | 1.07 ± 0.19 (Feb) | 0.68 ± 0.24 (Mar) |
| IASBS | 36.71 | 48.51 | 1805 | 2010–2012 | 0.20 ± 0.15 | 0.30 ± 0.22 (May) | 0.081 ± 0.036 (Dec) | 0.90 ± 0.47 | 1.59 ± 0.27 (Dec) | 0.41 ± 0.21 (Jun) |
| **Blida** | 36.51 | 2.88 | 230 | 2004–2010 | 0.22 ± 0.17 | 0.36 ± 0.18 (Jul) | 0.11 ± 0.07 (Jul) | 0.92 ± 0.40 | 1.10 ± 0.37 (Jan) | 0.72 ± 0.38 (Jul) |
| **Lampedusa** | 35.52 | 12.63 | 45 | 2000–2012 | 0.18 ± 0.15 | 0.24 ± 0.14 (Jul) | 0.085 ± 0.050 (Nov) | 0.91 ± 0.50 | 1.08 ± 0.54 (Aug) | 0.55 ± 0.28 (Dec) |
| Ras El Ain | 31.67 | -7.60 | 570 | 2006–2007 | 0.24 ± 0.18 | 0.46 ± 0.22 (Jul) | 0.090 ± 0.052 (Feb) | 0.74 ± 0.38 | 1.15 ± 0.36 (Apr) | 0.35 ± 0.19 (Jul) |
| Saada | 31.63 | -8.16 | 420 | 2004–2012 | 0.22 ± 0.17 | 0.39 ± 0.23 (Jul) | 0.087 ± 0.050 (Jan) | 0.73 ± 0.38 | 1.00 ± 0.38 (Dec) | 0.48 ± 0.27 (Jul) |
| Ouarzazate | 30.93 | -6.91 | 1136 | 2012 | 0.16 ± 0.18 | 0.38 ± 0.22 (Aug) | 0.033 ± 0.016 (Dec) | 0.49 ± 0.32 | 0.96 ± 0.26 (Dec) | 0.17 ± 0.11 (Jul) |
| **Sede Boker** | 30.86 | 34.78 | 480 | 1999–2012 | 0.18 ± 0.13 | 0.26 ± 0.17 (Apr) | 0.11 ± 0.08 (Dec) | 0.94 ± 0.44 | 1.18 ± 0.29 (Aug) | 0.57 ± 0.40 (Apr) |
| Eilat | 29.50 | 34.92 | 15 | 2007–2012 | 0.20 ± 0.15 | 0.29 ± 0.21 (Apr) | 0.11 ± 0.04 (Jan) | 0.87 ± 0.41 | 1.20 ± 0.36 (Jul) | 0.56 ± 0.38 (Apr) |
| La Laguna | 28.48 | -16.32 | 568 | 2006–2012 | 0.15 ± 0.16 | 0.28 ± 0.21 (Jul) | 0.055 ± 0.021 (Dec) | 0.61 ± 0.36 | 0.95 ± 0.46 (Dec) | 0.37 ± 0.24 (Aug) |
| Santa Cruz Tenerife | 28.47 | -16.25 | 52 | 2005–2012 | 0.16 ± 0.16 | 0.26 ± 0.20 (Jul) | 0.065 ± 0.028 (Dec) | 0.72 ± 0.41 | 0.90 ± 0.52 (Apr) | 0.54 ± 0.45 (Jul) |
| **Izana** | 28.31 | -16.50 | 2391 | 1999–2012 | 0.06 ± 0.1 | 0.15 ± 0.16 (Jul) | 0.015 ± 0.007 (Feb) | 0.97 ± 0.52 | 1.34 ± 0.37 (Dec) | 0.54 ± 0.50 (Aug) |
| **Dhadnah** | 25.51 | 56.33 | 81 | 2004–2010 | 0.37 ± 0.21 | 0.69 ± 0.20 (Jul) | 0.19 ± 0.10 (Jan) | 0.75 ± 0.42 | 1.20 ± 0.42 (Dec) | 0.44 ± 0.21 (Apr) |
| **Solar Village** | 24.91 | 46.40 | 764 | 1999–2012 | 0.35 ± 0.24 | 0.55 ± 0.32 (May) | 0.17 ± 0.14 (Jan) | 0.54 ± 0.35 | 0.83 ± 0.36 (Dec) | 0.22 ± 0.15 (May) |
| **Dahkla** | 23.72 | -15.95 | 12 | 2002–2003 | 0.30 ± 0.29 | 0.62 ± 0.34 (Jun) | 0.12 ± 0.05 (Dec) | 0.53 ± 0.34 | 0.73 ± 0.36 (Nov) | 0.30 ± 0.20 (Jul) |
| **Mezaira** | 23.15 | 53.78 | 204 | 2004–2012 | 0.35 ± 0.22 | 0.58 ± 0.21 (Jun) | 0.19 ± 0.07 (Dec) | 0.70 ± 0.41 | 1.10 ± 0.33 (Nov) | 0.30 ± 0.22 (Mar) |
| **Hamim** | 22.97 | 54.30 | 209 | 2004–2007 | 0.34 ± 0.20 | 0.58 ± 0.28 (Jun) | 0.18 ± 0.09 (Jan) | 0.67 ± 0.41 | 1.22 ± 0.46 (Dec) | 0.27 ± 0.17 (Jun) |
| **Tamanrasset INM** | 22.79 | 5.53 | 1377 | 2006–2012 | 0.21 ± 0.25 | 0.39 ± 0.35 (Aug) | 0.056 ± 0.045 (Jan) | 0.51 ± 0.32 | 0.80 ± 0.32 (Jan) | 0.20 ± 0.14 (Jun) |
| **KAUST** | 22.31 | 39.10 | 11 | 2012 | 0.49 ± 0.45 | 0.67 ± 0.81 (Mar) | 0.36 ± 0.19 (Apr) | 0.76 ± 0.37 | 1.24 ± 0.28 (Nov) | 0.40 ± 0.17 (May) |
| Agoufou | 15.35 | -1.48 | 305 | 2003–2009 | 0.51 ± 0.41 | 0.77 ± 0.41 (Jun) | 0.28 ± 0.24 (Dec) | 0.29 ± 0.23 | 0.53 ± 0.25 (Dec) | 0.092 ± 0.099 (Jun) |
| **Dakar** | 14.39 | -16.96 | 0 | 1999–2012 | 0.44 ± 0.30 | 0.62 ± 0.29 (Jun) | 0.30 ± 0.19 (Nov) | 0.36 ± 0.25 | 0.62 ± 0.30 (Dec) | 0.19 ± 0.15 (Jun) |
| Zinder Airport | 13.78 | 8.99 | 456 | 2009–2012 | 0.52 ± 0.41 | 0.89 ± 0.56 (May) | 0.32 ± 0.28 (Dec) | 0.35 ± 0.25 | 0.51 ± 0.29 (Dec) | 0.14 ± 0.11 (May) |
| **Banizoumbou** | 13.54 | 2.67 | 250 | 1999–2012 | 0.52 ± 0.42 | 0.89 ± 0.57 (Mar) | 0.29 ± 0.23 (Dec) | 0.35 ± 0.25 | 0.54 ± 0.29 (Dec) | 0.16 ± 0.21 (Jun) |
| **DMN Maine Soroa** | 13.22 | 12.02 | 350 | 2005–2010 | 0.48 ± 0.39 | 1.01 ± 0.75 (May) | 0.26 ± 0.14 (Dec) | 0.37 ± 0.30 | 0.62 ± 0.35 (Dec) | 0.10 ± 0.09 (Jun) |
| **Ouagadougou** | 12.20 | -1.40 | 290 | 1999–2007 | 0.52 ± 0.44 | 0.88 ± 0.70 (Mar) | 0.33 ± 0.28 (Dec) | 0.40 ± 0.24 | 0.56 ± 0.26 (Dec) | 0.24 ± 0.11 (Mar) |
| **Djougou** | 9.76 | 1.60 | 400 | 2004–2007 | 0.66 ± 0.44 | 0.97 ± 0.48 (Mar) | 0.35 ± 0.14 (Oct) | 0.52 ± 0.34 | 0.96 ± 0.30 (Dec) | 0.27 ± 0.12 (Mar) |
| **Ilorin** | 8.32 | 4.34 | 350 | 1999–2012 | 0.67 ± 0.49 | 1.10 ± 0.56 (Feb) | 0.38 ± 0.22 (Jun) | 0.66 ± 0.36 | 0.91 ± 0.30 (Dec) | 0.33 ± 0.16 (Apr) |
| **Ascension Island** | -7.98 | -14.42 | 30 | 1999–2012 | 0.16 ± 0.10 | 0.32 ± 0.14 (Sep) | 0.086 ± 0.057 (Nov) | 0.70 ± 0.37 | 1.34 ± 0.17 (Sep) | 0.280 ± 0.147 (Apr) |
| **Mongu** | -15.25 | 23.15 | 1107 | 1999–2009 | 0.21 ± 0.19 | 0.50 ± 0.26 (Sep) | 0.080 ± 0.040 (Apr) | 1.60 ± 0.43 | 1.85 ± 0.16 (Aug) | 0.812 ± 0.363 (Jan) |
| Etosha Pan | -19.18 | 15.91 | 1131 | 2000–2001 | 0.15 ± 0.15 | 0.40 ± 0.17 (Oct) | 0.069 ± 0.042 (May) | 1.44 ± 0.43 | 1.80 ± 0.16 (Oct) | 1.14 ± 0.40 (Nov) |
| Reunion St. Dénis | -20.88 | 31.59 | 150 | 2007–2012 | 0.064 ± 0.036 | 0.095 ± 0.044 (Oct) | 0.046 ± 0.018 (Jul) | 0.70 ± 0.36 | 1.12 ± 0.28 (Oct) | 0.452 ± 0.270 (Jul) |
| Skukuza | -24.99 | 31.59 | 150 | 1999–2011 | 0.18 ± 0.14 | 0.27 ± 0.18 (Sep) | 0.13 ± 0.09 (Jul) | 1.34 ± 0.42 | 1.46 ± 0.28 (Sep) | 0.996 ± 0.473 (Jan) |

Region groupings (left margin): *Northern African and Middle Eastern Sites* (Granada through KAUST); *Western Africa Sites* (Agoufou through Ilorin); *Southern Africa* (Ascension Island through Skukuza).

[a] The month of the reported max/min value is indicated in parenthesis




**Table 2.** Global and Africa-only annual average burdens, lifetimes, total deposition fluxes, and fraction wet deposition of four prognostic aerosol species in CCAM for the year 2010.

| Species | Burden (Tg) | | Total deposition (Tg a$^{-1}$) | | Fraction wet deposition of total | | Lifetime (days) | | Emissions (Tg yr$^{-1}$) | |
|---|---|---|---|---|---|---|---|---|---|---|
| | Global | Africa | Global | Africa | Global | Africa | Global | Africa | Global | Africa |
| **BC** | 0.187 | 0.0465 | 6.84 | 1.56 | 0.844 | 0.802 | 9.98 | 10.9 | 7.38 | 2.05 |
| **OC** | 1.11 | 0.305 | 44.1 | 12.1 | 0.819 | 0.782 | 9.22 | 9.19 | 44.8 | 14.8 |
| **Sulfate** | 0.961 | 0.161 | 65.1 | 7.18 | 0.865 | 0.833 | 5.39 | 8.16 | 57 | 9.18 |
| **Dust** | 67.7 | 26.9 | 2780 | 1460 | 0.565 | 0.364 | 8.9 | 6.72 | 2805 | 2320 |





**Table 3. Summary of model-observation comparison of monthly-average AOD$_{550nm}$. The significance of the Pearson's correlation is indicated by '\*' for p<0.05, '\*\*' for p<0.01, and '\*\*\*' for p<0.001; NS is not significant at 0.05 level.**

| Site | Correlation coefficient (r) | | Normalized Mean Bias | Mean Absolute Error | Number of months |
|------|------|------|------|------|------|
| Granada | 0.47 | *** | 176.6 % | 0.27 | 50 |
| Blida | 0.70 | *** | 220.0 % | 0.54 | 33 |
| Lampedusa | 0.58 | *** | 278.2 % | 0.51 | 46 |
| Saada | 0.60 | ** | 231.7 % | 0.50 | 74 |
| Sede_Boker | 0.23 | * | 245.5% | 0.43 | 129 |
| Izana | 0.46 | ** | 970.0 % | 0.65 | 75 |
| Dhadnah | 0.81 | *** | 125.1 % | 0.45 | 50 |
| Solar Village | 0.51 | *** | 121.1 % | 0.42 | 128 |
| Dahkla | 0.49 | * | 242.2 % | 0.75 | 19 |
| Hamim | 0.82 | *** | 115.2 % | 0.37 | 28 |
| Tamanrasset_INM | 0.89 | ** | 253.6 % | 0.51 | 19 |
| Agoufou | 0.51 | ** | 89.7 % | 0.47 | 58 |
| Dakar | 0.33 | ** | 103.2 % | 0.48 | 95 |
| Zinder_Airport | 0.61 | ** | 59.3 % | 0.35 | 30 |
| Banizoumbou | 0.50 | ** | 58.6 % | 0.34 | 126 |
| DMN_Maine_Soroa | 0.52 | ** | 94.5 % | 0.46 | 41 |
| Ouagadougou | 0.27 | * | 29.3 % | 0.28 | 61 |
| Djougou | 0.29 | NS | -1.3 % | 0.20 | 24 |
| Ilorin | 0.59 | ** | -12.6 % | 0.22 | 61 |
| Ascension_Island | 0.51 | ** | 41.8 % | 0.09 | 53 |
| Mongu | 0.67 | ** | -21.2 % | 0.09 | 77 |
| Reunion – St Dénis | 0.21 | NS | 135.0 % | 0.09 | 84 |
| Skukuza | 0.48 | ** | 24.6 % | 0.07 | 72 |





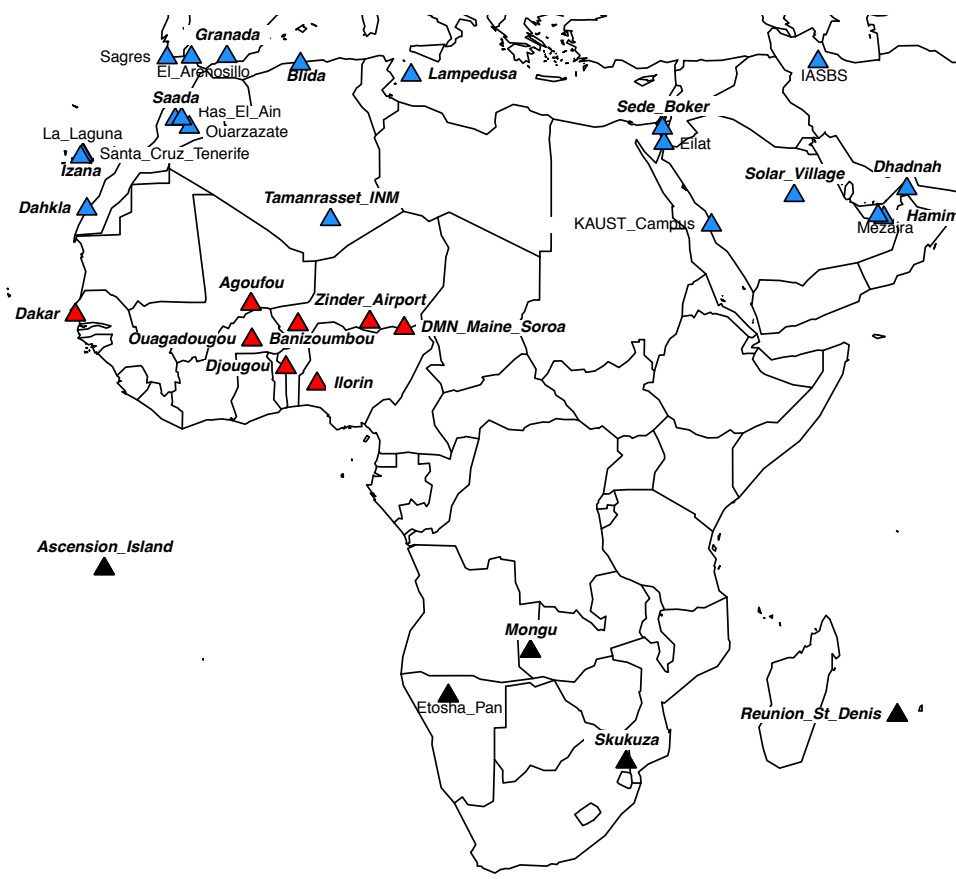

**Figure 1: Map of long-term AERONET sites used in this study. Sites are color-coded by general geographic area and aerosol source type. Site names in bold italics are used in model comparison.**





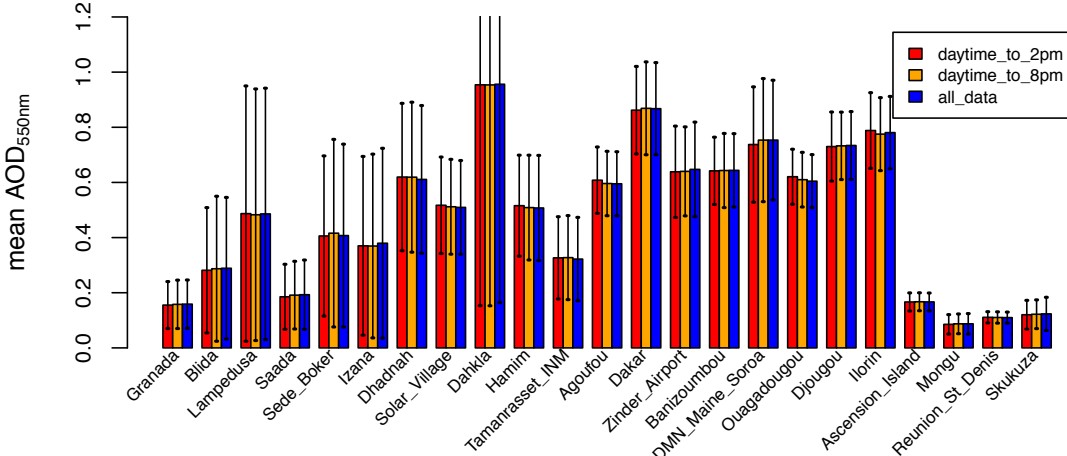

**Figure 2: Comparison of methods to compute mean modeled AOD$_{550nm}$, for an example in January 2000: red bars include model output only for 6am to 2pm local time; yellow bars for 6am to 8pm local time; and blue bars for 24 hours.**

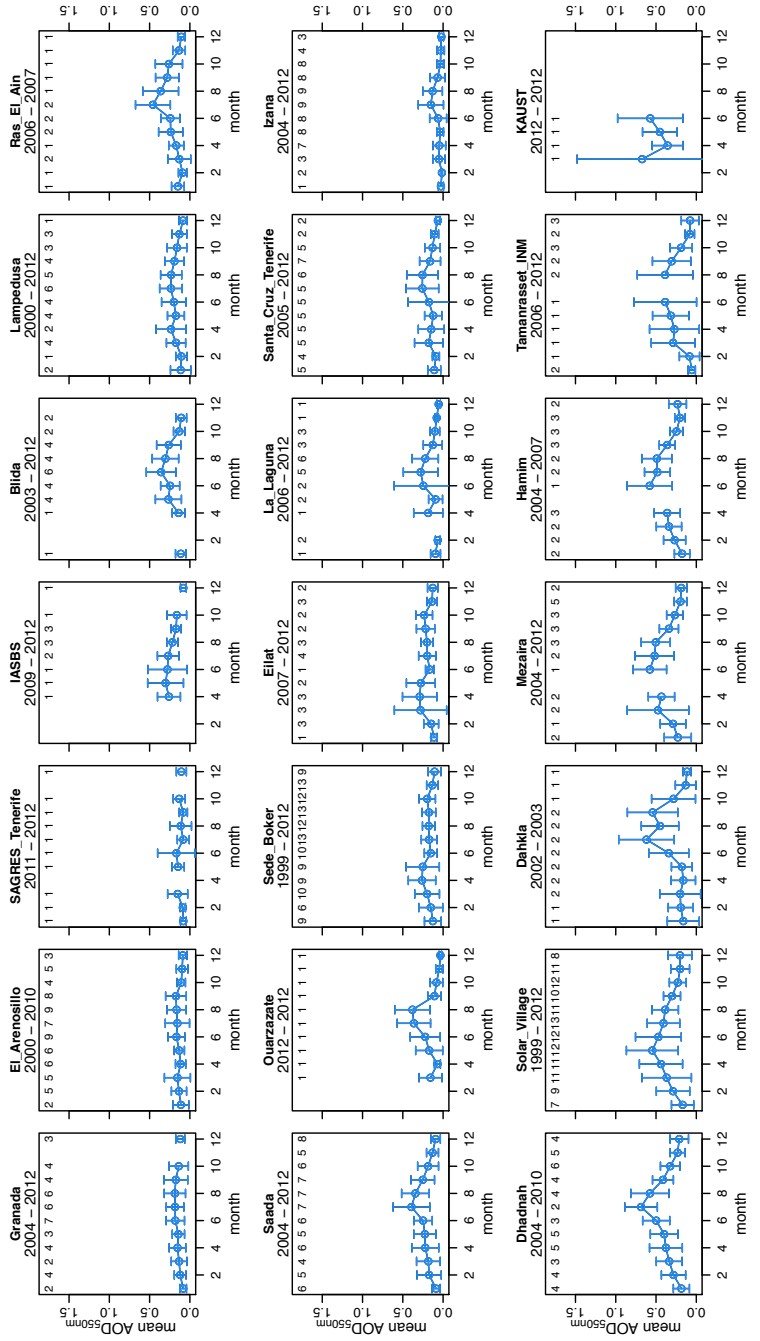

**Figure 3a: Multi-year mean seasonal cycle of AOD$_{550nm}$ at long-term AERONET sites in Northern Africa and the Middle East. The number of years of data used for each month is shown at the top of the plot area, and the total range of years of observations used is listed under each site name.**





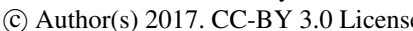

**Figure 3b:** Multi-year mean seasonal cycle of $AOD_{550nm}$ at long-term AERONET sites in Western and Southern Africa. The number of years of data used for each month is shown at the top of the plot area, and the total range of years of observations used is listed under each site name.





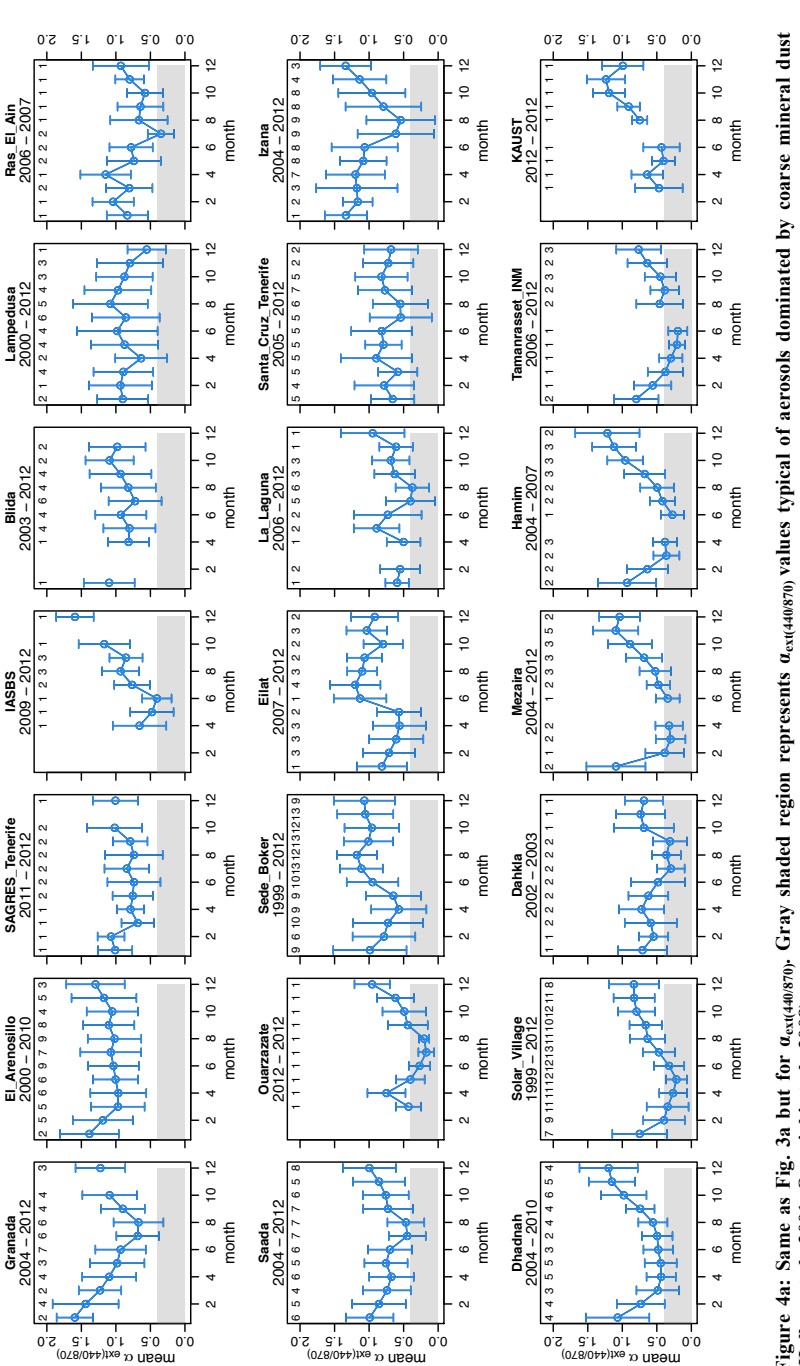

**Figure 4a: Same as Fig. 3a but for** $\alpha_{ext(440/870)}$**. Gray shaded region represents** $\alpha_{ext(440/870)}$ **values typical of aerosols dominated by coarse mineral dust (Holben et al., 2001; Ogunjobi et al., 2008).**





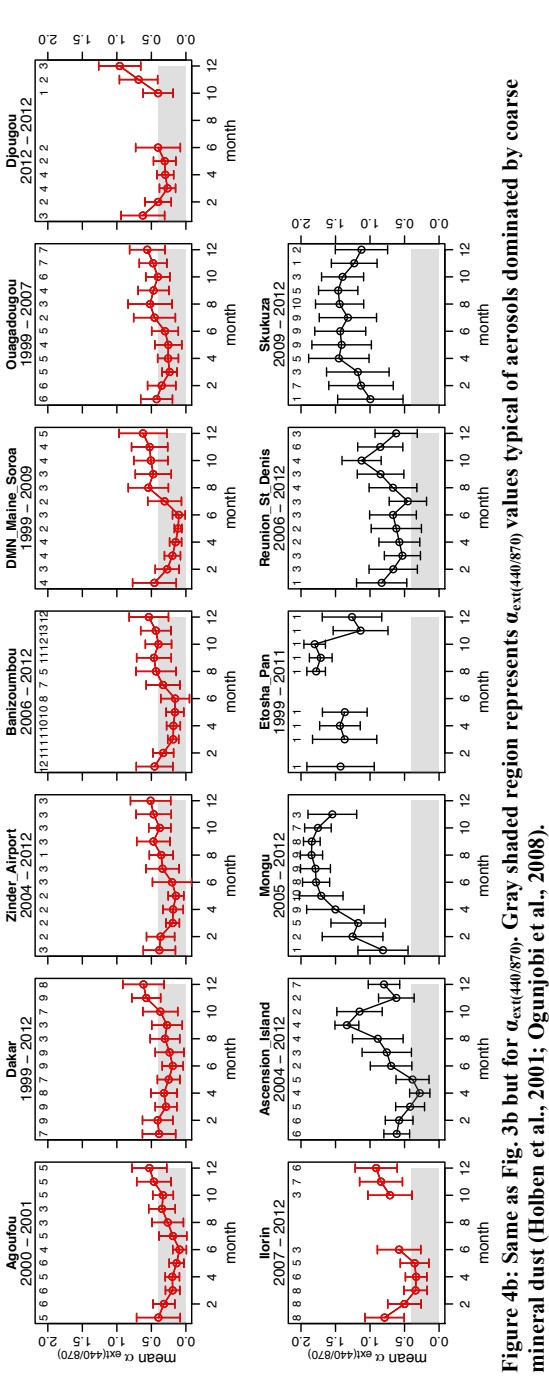

**Figure 4b: Same as Fig. 3b but for** $\alpha_{ext(440/870)}$**. Gray shaded region represents** $\alpha_{ext(440/870)}$ **values typical of aerosols dominated by coarse mineral dust (Holben et al., 2001; Ogunjobi et al., 2008).**





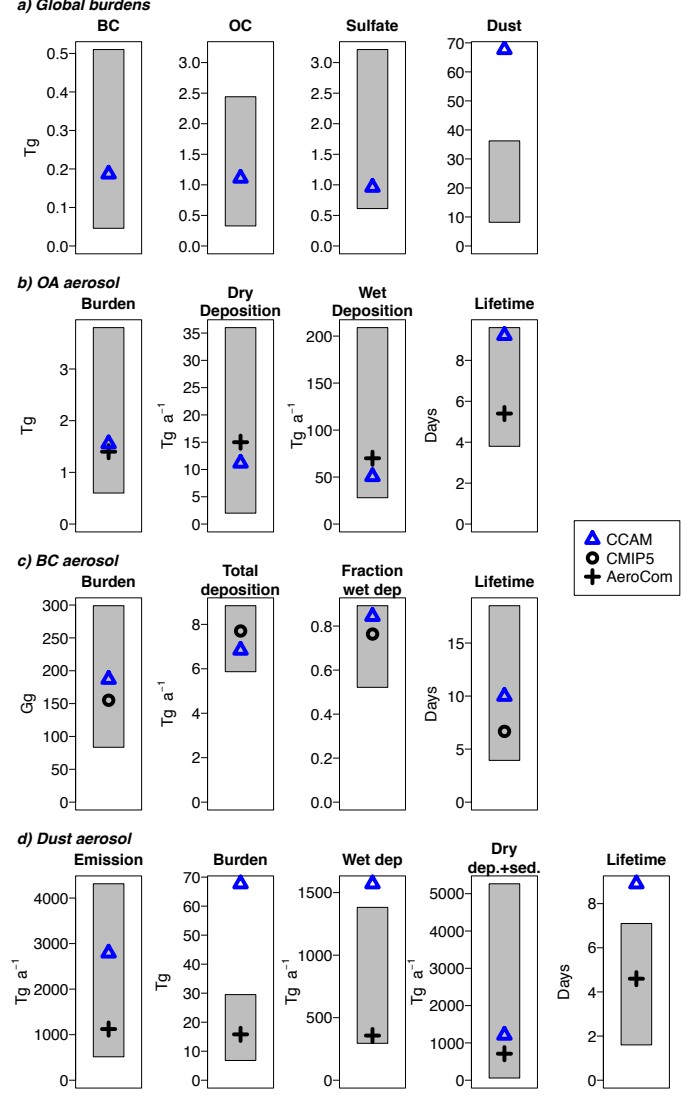

**Figure 5: Comparison of present-day model results for CCAM (blue triangles) against ranges from other models (shaded gray area), for a) global burdens of major aerosol constituents, b) characteristics of OA aerosol, c) characteristics of BC aerosol, and d) characteristic of dust aerosol. Reference model ranges in a) are from Kinne et al. (2006) with additional models provided from Jathar et al. (2011) for OC, Liu et al. (2005) for sulfate, and Zender et al. (2004) for dust. AeroCom Phase II model ranges and medians (black crosses) in b) are from Tsigaridis et al. (2014); CCAM modeled OC is converted to OA by multiplying by a factor of 1.4 for a consistent comparison (Tsigaridis et al., 2014). CMIP5 model ranges and medians (black circles) in c) are from Allen and Landuyt (2014). AeroCom Phase I model ranges and medians (black crosses) in d) are from Huneeus et al. (2011).**





**Figure 6: Multi-year mean seasonal cycle of AOD$_{550nm}$ for observed (red) and modeled with all CCAM outputs, 1999–2012 (blue), and only those months with AERONET data meeting the 70% completeness cutoff (yellow). ± 1 standard deviation for the observations and CCAM 1999–2012 output is shaded.**



**Figure 6, continued: Multi-year mean seasonal cycle of AOD$_{550nm}$ for observed (red) and modeled with all CCAM outputs, 1999–2012 (blue), and only those months with AERONET data meeting the 70% completeness cutoff (yellow). ± 1 standard deviation for the observations and CCAM 1999–2012 output is shaded.**





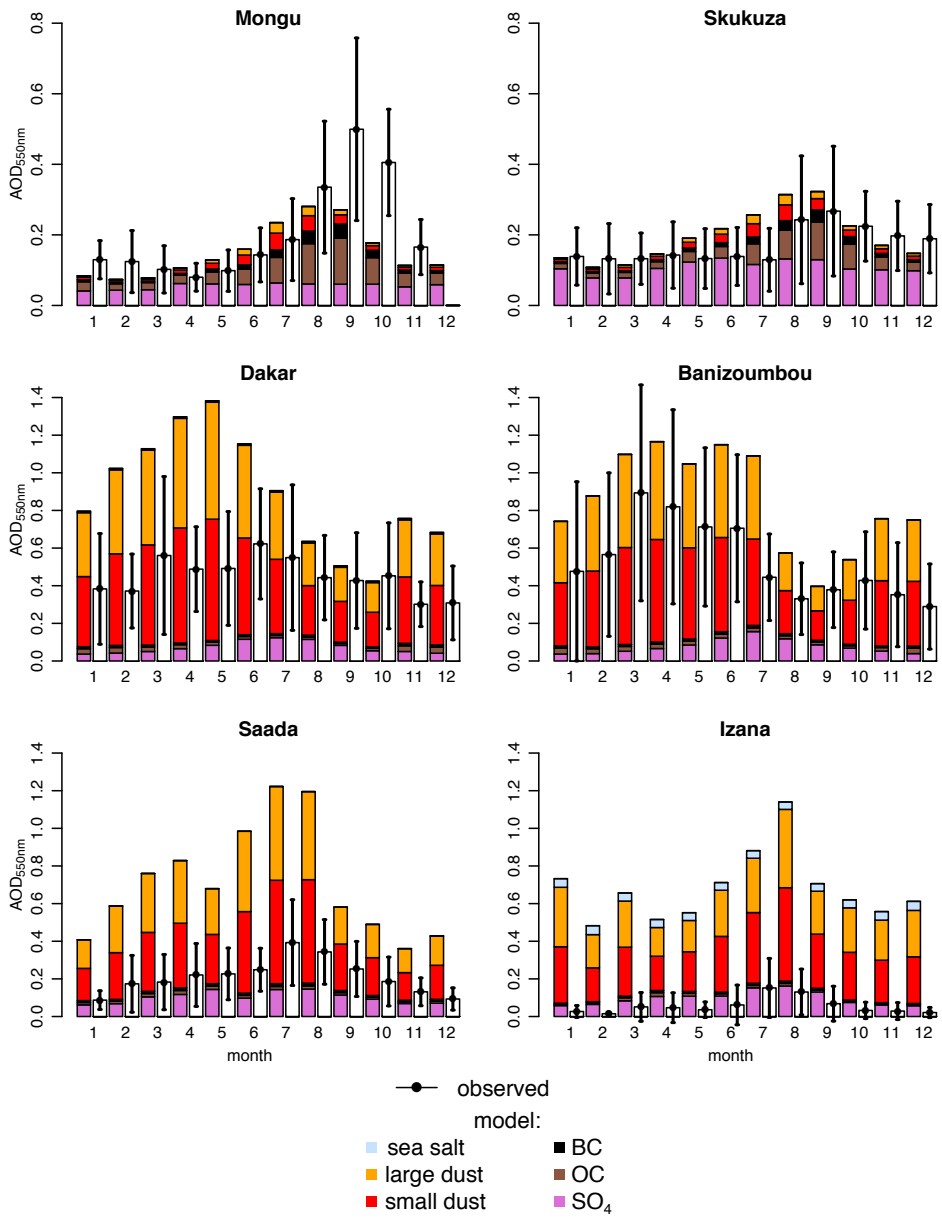

Figure 7: Multi-year mean observed vs. modeled seasonal cycle of $AOD_{550nmnm}$ at six AERONET sites. Modeled $AOD_{550nm}$ is broken down into the contribution from each aerosol species (sea salt, large size bin dust (radius $\geq$ 1 μm), small size bin dust (radius < 1 μm), BC, OC, sulfate ($SO_4$)).





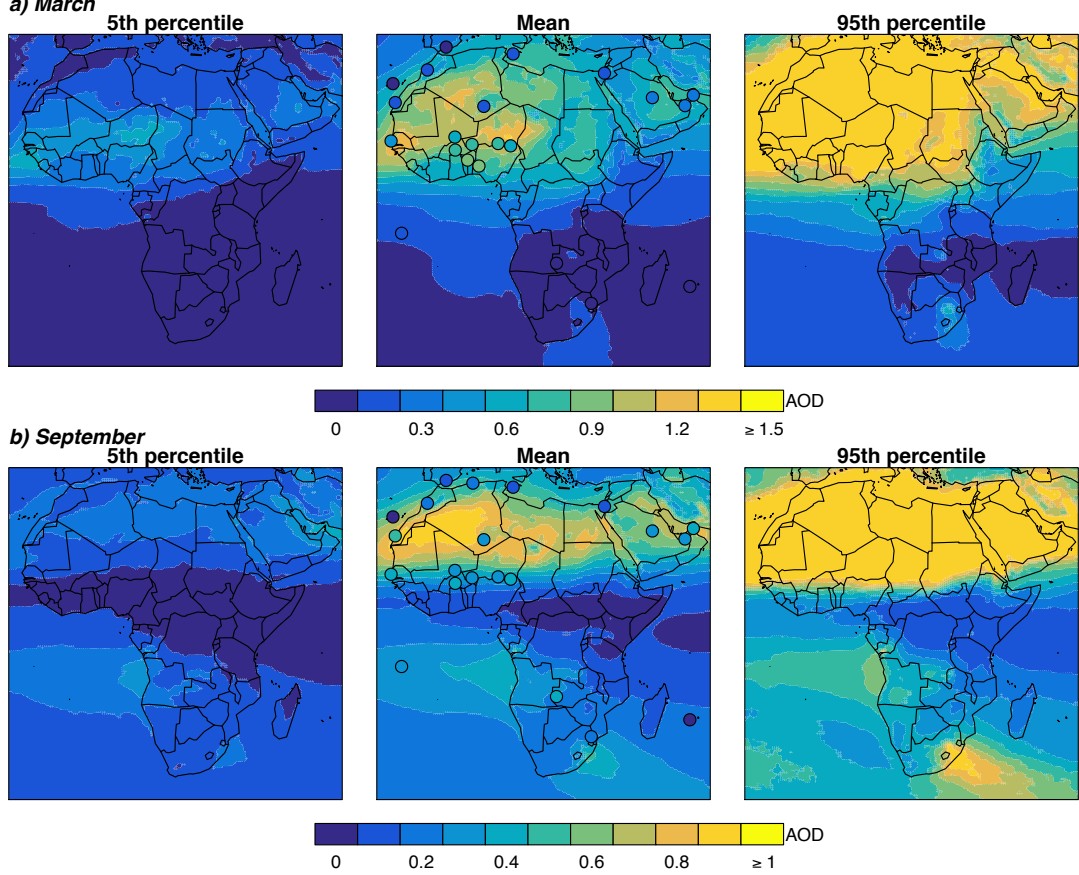

**Figure 8: Seasonal variation in multi-year monthly 5$^{th}$ percentile, mean, and 95$^{th}$ percentile modeled AOD$_{550nm}$ (map background) for the full model climatology (1999–2012), with observed multi-year means (points) for all available AERONET data, for a) September and b) March. The number of years and range of years used for each site is the same as in Fig. 3.**