# Peer review of "Evaluation of climate model aerosol seasonal and spatial variability over Africa using AERONET"

_Atmospheric Chemistry and Physics, 2017_

## Referee Comment (RC1) · Anonymous Referee #1 · 23 May 2017

This paper evaluates the performance of the CCAM model at simulating aerosols over Africa, by comparison to AERONET data. The paper's title and some of the text set it up to be primarily a description of the aerosol cycle in Africa. However most of the real content is in the evaluation against AERONET, where we see that there are some shortcomings for CCAM's representation of dust. As a result, I don't think it makes sense to present this as a paper about the seasonality of aerosols in Africa. It's really a model evaluation exercise, which establishes some problems with dust and the timing of biomass burning, but better performance for other aerosols. So perhaps there will be a follow up in a few years when these issues have been improved and the model is

more in the application phase than the evaluation phase. As a result this paper might fit better thematically in GMD than in ACP, but it is within scope for ACP as well.

The paper is interesting and scientifically does not have major problems. However, the organization should be improved. There are parts where it is a bit lengthy and unclear, and contains statements which are either slightly incorrect or information that is not necessary (it reads as very descriptive and not very analytical, sometimes, if that makes sense). This makes it difficult to read and pick out the main points. The whole paper could be streamlined to improve readability and clarity. I have included some suggestions for where to do this in my comments below. These rewrites should make it easier to judge the paper and pull out the main conclusions, which I have a bit of a hard time doing now. As a result I recommend major revisions since some of the suggested rewrites will alter the structure of the paper somewhat and some things may become clearer. I would like to review the revised version.

Title: See above comments. I recommend changing the title to make clear that the focus is the evaluation of the model, rather than "Understanding the seasonality and climatology of aerosols in Africa".

Abstract: This should ideally be one paragraph which concisely summarises the key points of the paper. This abstract is three long paragraphs covering about a page. I suggest that this can be condensed somewhat. For example, the entire middle paragraph is more or less well-known results (e.g. where and when dust comes from) and can be deleted. I would then merge the remaining two paragraphs, which contain more overview and then the main results of the paper.

Section 2.1: In my Quick Report comments I had suggested adding more AERONET sites; the authors added most of these (thank you for this effort), but not one of the key Saharan dust outflow sites which I had suggested (Capo Verde). I see that this is just outside of model domain listed here in the paper, so perhaps that is why it is not included. But presumably the model was run globally so perhaps the analysis domain

could be extended another few degrees to include this site? It is one of the key long-term sites which has been used by many researchers to examine Saharan dust and evaluate models (among other things) so would be useful to have the comparison there as a point of reference, if possible. While not essential, I mention this specific site again for this reason. It could help confirm the hypothesis about dust lifetime in CCAM, since this site is a way away from the sources.

Izana is not a useful site for model evaluation and can be removed. It is on the top of a mountain and not representative of the surrounding area. See e.g. https://aeronet.gsfc.nasa.gov/new_web/photo_db/Izana.html .

Page 5 line 26: Strictly AERONET does not measure AOD. It measures the direct solar irradiance, and then does a (very accurate) retrieval to determine AOD. Even this direct-Sun AOD product is a retrieval, not a direct measurement. Also, the wavelength range given here is wrong (the range depends on the specific instrument). I suggest rewording to say that AERONET provides spectral AOD at multiple wavelengths, depending on instrument, from the UV to the swIR. A key point being changing the word "measured" here and in line 28 (plus other places I might have missed) to a more correct term such as "provides".

Equation 2: The definition of AOD seems superfluous here so can probably be deleted as assumed background knowledge.

Page 6 lines 30-31: "was considered, and were aligned as possible" does not make sense. I suggest rewording this paragraph (perhaps it is just this first sentence which is causing confusion). If I understand correctly then the model provides 6-hourly output and a daily average was constructed from the output from 'daytime' hours over this domain. The key point being here is that sampling is daytime only to match AERONET, but the specific AERONET days are not being matched directly. Is that correct?

Page 7, lines 12: Likewise, I think the definition of Pearson correlation coefficient is not necessary. For the specific analyses performed in the paper (i.e. assessing to

what extent the seasonality of AERONET is reproduced by CCAM), the coefficient of determination (r2) may be useful than r anyway.

Page 7, lines 22-23: since AOD distributions are not Gaussian, might it be better to show interquartile range or similar rather than standard deviation?

Page 7, lines 24-25: This is another example of a slightly misleading/inaccurate statement. Ångström exponent (AE) is related to the optical dominance of fine vs. coarse aerosols in the column. This is subtly different from what is written in the paper which says that it gives information on size. For example, an AE around 1 could be either an indicator of monomodal mid-sized aerosols, or an indicator of a column containing similar amounts (in optical terms) of fine and coarse aerosols. These are quite different things. I suggest rewording.

Page 8, lines 2-3: "regional trends". It would be better to say "regional patterns" or something, since the term "trend" is most commonly used to refer to analyses of time series for changes.

Section 3: I don't think that the general description of aerosol seasonality in the model is that necessary, since the main aerosol sources in Africa and their timings are reasonably well-known, and the model has some biases anyway. (Really, the evaluation should have come before this descriptive section anyway, since you have to establish the validity of the model before you can use it to answer science questions.) It would be better in my view to present and discuss model and AERONET seasonality for each region simultaneously. Then we can get to the interesting stuff of whether the model is reproducing the patterns seen in AERONET. Essentially, merge in the current Section 4.2 with the existing Section 3 and rewrite.

Section 3.1: as an example of some stylistic issues throughout the paper (applicable to much of the discussion, not just here): 1. The word "values" appears a lot here and can probably be deleted. There isn't a real difference between saying "the AOD values" or just saying "the AOD", for example, and the latter is more concise and readable.

2. Similarly, the subscripts for AOD and AE are the same all the time so can be omitted for brevity and clarity. (For example, just say once at the start of the data set description the wavelength or wavelength range being considered and don't repeat it every time).

3. The text in this section also doesn't specify whether AERONET or model data are being referred to. The related Figure 3 caption also doesn't say. This should be listed explicitly. I infer it is the model.

Page 14, line 5: As another style example, "The Pearson's correlation coefficient" could have "The" and probably "Pearson's" deleted as well.

Page 14, line 16: is the beta here intentional? If so, what does it mean?

Figure 3: In general I don't see the point of these figures. Seeing one line per site here is not very informative. If the purpose of the paper is to compare with AERONET, the same basic information for AOD is repeated in Figure 6. Or am I misunderstanding something? It would be better to show, for each site, the model and AERONET together so a direct comparison can be made. So something like Figure 6, for both AOD and AE.

Figure 7: It would be better to overplot the AERONET AOD on top of the model component lines, rather than shifting it off to the right, to allow a more clear visual comparison of aerosol amount and seasonality.

Table 1: It would be useful to perform the AERONET/model comparison at ALL the sites shown, not just a subset. Otherwise what is the point of including them in the paper if the AERONET data are not used?

Table 3: I am not sure it is useful to report significance of correlation coefficients here. I don't think that it adds anything to the analysis or discussion, and due to strong autocorrelation of the data (which I don't think is accounted for) it is possible that the significance estimates are incorrect anyway.

General: as noted in my Quick Report, I suggest the authors also perform some analysis using daily (rather than monthly) data. This can be simple visual scatter plots for each site, or something similar to Table 3. This will help to tell to what extent biases in the monthly data are due to aerosol events that are missed in the model, and to what extent they are systematic biases in component loadings or optical properties. Going to daily data here also helps to avoid some of the sampling differences.

---

## Referee Comment (RC2) · Anonymous Referee #2 · 29 May 2017

General comments: In this study the regional and seasonal representation of aerosols in the global CCAM model is evaluated for the African domain, mainly through comparison of modeled and ground based retrievals of AOD from AERONET across Africa and parts of the Middle East.

I find the paper scientifically interesting and mostly well written, and the presentation

of model and observationally based results should be useful for others planning to do similar model evaluation studies. Parts of the model description is vague, however, which makes it difficult for the reader to find necessary information about the aerosol treatment without actually reading many of the underlying papers for the model. The treatment of sea salt, in particular, is poorly described, and the potential impact of this component (on coastal and island sites) on the results has mainly been omitted, except (for some sites) in Figure 7.

Specific comments and technical corrections: (For simplicity, the arrow symbol "->" is used to suggest a change from text version A -> B)

Page 1, line 18: "ground-based observations" should be changed to "ground-based remote retrievals" or something along that line.

Page 3, line 8: "may also feedback on climate" -> "may also feed back on climate" (feedback is a noun)

Page 3, line 15: "first AeroCom" -> "first phase of AeroCom"

Page 3, line 30 and throughout Sect. 2.1: What does non-prognostic / diagnostic sea salt aerosols mean? Do you prescribe the emissions or the concentrations? A reference for this treatment should be added, e.g. after the additional (but not sufficient) info on page 5, line 3.

Page 4, line 12; "spun-up" -> "spun up"

Page 4, line 19: Is also the semi-direct effect taken into account?

Page 4, line 31: The "-" in "-2" in the exponent (m**-2) is misplaced.

Page 5, line 13: "vary every 5 years" "they vary every 5 years"

Page 5, line 14: "anthropogenic" -> "non-biomass burning anthropogenic"

Page 5, line 17: "found a chemical" -> "found that a chemical"

[Figure]

Page 5, line 17: Unless sea salt concentrations are prescribed, why are these large particles not also affected by gravitational settling?

Page 6, line 2: "AOD" -> "AOD at 550 nm" P age 6, line 5: "34 sites Africa" -> "34 sites in Africa"

Page 6, line 18: "bolded site names" -> "site names in bold font"

Page 6, lines 24-25: I would suggest to rewrite "where if more than 30% of the daily values were missing, a monthly average could not be calculated " to ". I.e., if more than 30% of the daily values were missing, a monthly average was not calculated ".

Page 6. line 27: "This is to ensure the" -> "This is to ensure that the"

Page 6, lines 30 and 32: The sentence containing "and were aligned as possible" does not make sense, and the meaning of the following sentence is not clear to me either: Should it read "assessed for the averaging period in question" or "assessed for the respective averaging period"?

Page 7, lines 5-9: Unclear description of the 2 different calculations: "2) all model years" does not preclude 1). Should it be "2) all months of all model years"?

Page 7: Eqs. 4 and 5 are well known and can be skipped, or replaced with an equation for r. Line 17 also repeats the info on line 15.

Page 7, line 25: The Angstrom parameter does not equal 2 for all sub-micron particles. It is more correct to write "very fine particles" or "predominantly fine particles", or something along that line.

Page 7, line 27 (and throughout the manuscript): Small Angstrom parameter values can also be due to aerosols dominated by coarse sea salt aerosols. Perhaps this is not the case for this particular model and the sites studied here, but this should somehow be shown, at least for the coastal and island sites.

Page 9, line 4: As above.

Page 10, line 31: "fraction wet deposition" -> "wet deposition fraction"

Pages 10-11: Whether this aerosol is prescribed or not, sea salt influences the total AOD and should be discussed and included in Table 2, and also in Fig. 5 if prescribed concentrations have not been used. The sea salt AOD values for coastal/island sites in Fig. 7 look small compared to many of the available AerCom models (http://aerocom.met.no/cgi-bin/aerocom/surfobs_annualrs.pl).

Page 11, line 1: "BC are higher" -> "BC burden and lifetime are higher". Page 11, line 23: "areas are ±1 standard deviation" -> "areas are within ±1 standard deviation" (and the same for the following line).

Page 12, line 1: "spurious summertime peaks" -> "missing summertime peaks"

Page 12, lines 11-12: Since there is a severe overestimate in modeled AOD for some sites and some months, the sentence starting with "The model generally represents the magnitude of AOD550nm" should be rephrased.

Page 13, line 8: "The AOD550" -> "The observed AOD550" (?)

Page 13, line 15: "at the source" -> "at the biomass burning source"

Page 14, line 1: Is the ERA wind bias in winter consistent with the magnitude of the AOD bias? Could you make a simple estimate of this?

Page 14, line 16: There is a "beta" to much in "0.27 m s-1ß"

Page 14, line 31 and Page 16, lines 17-20: Can you show that the precipitation in the model is underestimated compared to observations (therefore explaining some of the positive dust emission bias)?

Page 16, line 31: "is slightly underestimated" -> "is underestimated"

Page 17, line 20: "CCAM is able to capture the general seasonal cycle of the emissions of dust, and the transport of all aerosol types". This has not been shown, and such a

[Figure]

statement should be limited to the aerosol components covered by the study.

Table 1: The second sentence in the table caption is grammatically incomplete.

Table 2: The gray shading should be explained in the table caption (as in the text).

Figure 1, caption: "used in model comparison" -> "used in the model comparison".

Figures 2-4, caption: Explain the whiskers.

---

## Author Comment (AC1) · 12 Sep 2017

The paper is interesting and scientifically does not have major problems. However, the organization should be improved. There are parts where it is a bit lengthy and unclear, and contains statements which are either slightly incorrect or information that is not necessary (it reads as very descriptive and not very analytical, sometimes, if that makes sense). This makes it difficult to read and pick out the main points. The whole paper could be streamlined to improve readability and clarity. I have included some suggestions for where to do this in my comments below. These rewrites should make it easier to judge the paper and pull out the main conclusions, which I have a bit of a hard time doing now. As a result I recommend major revisions since some of the suggested rewrites will alter the structure of the paper somewhat and some things may become clearer. I would like to review the revised version.

Title: See above comments. I recommend changing the title to make clear that the focus is the evaluation of the model, rather than "Understanding the seasonality and climatology of aerosols in Africa".

We revise the title to: "Evaluation of climate model aerosol seasonal and spatial variability over Africa using AERONET".

Abstract: This should ideally be one paragraph which concisely summarises the key points of the paper. This abstract is three long paragraphs covering about a page. I suggest that this can be

condensed somewhat. For example, the entire middle paragraph is more or less well-known results (e.g. where and when dust comes from) and can be deleted. I would then merge the remaining two paragraphs, which contain more overview and then the main results of the paper.

We condense the abstract following these recommendations.

Section 2.1: In my Quick Report comments I had suggested adding more AERONET sites; the authors added most of these (thank you for this effort), but not one of the key Saharan dust outflow sites which I had suggested (Capo Verde). I see that this is just outside of model domain listed here in the paper, so perhaps that is why it is not included. But presumably the model was run globally so perhaps the analysis domain could be extended another few degrees to include this site? It is one of the key long-term sites which has been used by many researchers to examine Saharan dust and evaluate models (among other things) so would be useful to have the comparison there as a point of reference, if possible. While not essential, I mention this specific site again for this reason. It could help confirm the hypothesis about dust lifetime in CCAM, since this site is a way away from the sources.1

The reviewer is correct that the African domain was extracted from global runs of CCAM. In those original runs, we would have been able to extract Cape Verde. We did in fact try to address this comment for this round of reviews; unfortunately, those original global runs were mistakenly deleted and therefore no longer available. We only have stored the African domain as in the current manuscript, and only for selected number of variables.

Izana is not a useful site for model evaluation and can be removed. It is on the top of a mountain and not representative of the surrounding area. See e.g. https://aeronet.gsfc.nasa.gov/new_web/photo_db/Izana.html .

We agree with the reviewer that the height of the observation site is an important consideration, which we include in Table 1 and discussed in the context of Izaña on page 8, lines 19-23 (Section 3.1). La Laguna, Santa Cruz Tenerife, and Izaña are in the same model grid box. In selecting sites for the observation-model comparisons where there were multiple sites in such close proximity, we originally selected sites that had the largest dataset available for the comparison in order to ensure the comparison was robust (which in this case was Izaña). In light of your comments and the differences in $\alpha_{ext}$ and the magnitude of AOD at Izaña, we instead evaluate the model with AERONET data from Santa Cruz, which has the second largest number of months with valid data available for comparison on the island. We revise Table 4, Figure 6, Figure 7, and the discussion and Section 4.2.3 accordingly.

Page 5 line 26: Strictly AERONET does not measure AOD. It measures the direct so- lar irradiance, and then does a (very accurate) retrieval to determine AOD. Even this direct-Sun AOD product is a retrieval, not a direct measurement. Also, the wavelength range given here is wrong (the range depends on the specific instrument). I suggest rewording to say that AERONET provides spectral AOD at multiple wavelengths, depending on instrument, from the UV to the swIR. A key point being changing the word "measured" here and in line 28 (plus other places I might have missed) to a more correct term such as "provides".

We revise the sentence in page 5 lines 25-26: "The global network of AERONET stations measure aerosol optical properties at multiple wavelengths ranging from the UV to shortwave infrared using a ground-based Cimel sun-photometer (Holben et al., 1998; Dubovik et al., 2002)."

we change "measured" throughout the manuscript.

Equation 2: The definition of AOD seems superfluous here so can probably be deleted as assumed background knowledge.

We remove the definition and equation 2.

Page 6 lines 30-31: "was considered, and were aligned as possible" does not make sense. I suggest rewording this paragraph (perhaps it is just this first sentence which is causing confusion). If I understand correctly then the model provides 6-hourly output and a daily average was constructed from the output from 'daytime' hours over this domain. The key point being here is that sampling is daytime only to match AERONET, but the specific AERONET days are not being matched directly. Is that correct?

This sentence has been removed and the entire paragraph clarified following your and the other reviewers' suggestions, as follows (page 6, lines 23-28): " Daily average AOD from AERONET is calculated for a minimum of 3 time points from sun photometer measurements, which can only be made during daytime, while modeled AOD is reported at 6-hourly resolution. Therefore, only CCAM AOD between 06:00 and 18:00 UTC was averaged for monthly and multi-year means (similar to other AERONET-model comparison studies; (e.g., Tegen et al., 2013). Model monthly means were, however, insensitive to the choice of daylight cut-off (see Fig. 2), which gives confidence that the instantaneous 6-hourly values from CCAM can represent the range of daytime hours sampled by AERONET."

Page 7, lines 12: Likewise, I think the definition of Pearson correlation coefficient is not necessary. For the specific analyses performed in the paper (i.e. assessing to what extent the seasonality of AERONET is reproduced by CCAM), the coefficient of determination (r2) may be useful than r anyway.

We have removed the definition.

Page 7, lines 22-23: since AOD distributions are not Gaussian, might it be better to show interquartile range or similar rather than standard deviation?

In order to address this comment we have split up the previous Table 1 into two tables. The new Table 1 has the average AOD and Ångström exponents with the standard deviation. We have also added the median, 25$^{th}$ and 75$^{th}$ percentile values to Table 1. We report averages and median values in the text as well. The timing of the maximum and minimum values has been moved to Table 2.

Page 7, lines 24-25: This is another example of a slightly misleading/inaccurate statement. Ångström exponent (AE) is related to the optical dominance of fine vs. coarse aerosols in the column. This is subtly different from what is written in the paper which says that it gives information on size. For example, an AE around 1 could be either an indicator of monomodal mid-sized aerosols, or an

indicator of a column containing similar amounts (in optical terms) of fine and coarse aerosols. These are quite different things. I suggest rewording.

Thank you for your help in clarifying our explanation of the use of the Angstrom exponent as a proxy related to aerosol size. This sentence has been updated following this suggestion and that of the other reviewer (now page 7, lines 16-18): "The Ångström exponent is an empirical proxy related to the relative contribution to optical thickness from coarse vs. fine aerosols, with values varying between approximately 0 for pure coarse dust particles to 2 for predominantly fine particles (Leon et al., 2009; Hamonou et al., 1999)."

Page 8, lines 2-3: "regional trends". It would be better to say "regional patterns" or something, since the term "trend" is most commonly used to refer to analyses of time series for changes.

We change "trends" to "patterns" as you suggest.

Section 3: I don't think that the general description of aerosol seasonality in the model is that necessary, since the main aerosol sources in Africa and their timings are reasonably well-known, and the model has some biases anyway. (Really, the evaluation should have come before this descriptive section anyway, since you have to establish the validity of the model before you can use it to answer science questions.) It would be better in my view to present and discuss model and AERONET seasonality for each region simultaneously. Then we can get to the interesting stuff of whether the model is reproducing the patterns seen in AERONET. Essentially, merge in the current Section 4.2 with the existing Section 3 and rewrite.

We agree with the reviewer that model evaluation should be performed prior to using the model to inform processes. Section 3 is not presenting model results, but rather the observational data from AERONET. The model evaluation is performed in Section 4, after the observational data are discussed. We felt it was beneficial to discuss the entire suite of available AERONET AOD at sites influenced by African dust and biomass burning independent of the model first. We feel that clarifications added throughout the paper to address the confusion resulting from misinterpreting Section 3 and Figures 3 and 4 as model results solves this issue.

Section 3.1: as an example of some stylistic issues throughout the paper (applicable to much of the discussion, not just here): 1. The word "values" appears a lot here and can probably be deleted. There isn't a real difference between saying "the AOD values" or just saying "the AOD", for example, and the latter is more concise and readable.

Values was removed throughout the paper wherever appropriate.

2. Similarly, the subscripts for AOD and AE are the same all the time so can be omitted for brevity and clarity. (For example, just say once at the start of the data set description the wavelength or wavelength range being considered and don't repeat it every time).

We retain the subscripts in the figures but remove from all text following the initial description in the methods section.

3. The text in this section also doesn't specify whether AERONET or model data are being referred to. The related Figure 3 caption also doesn't say. This should be listed explicitly. I infer it is the model.

We apologize that the title of this section and caption of Figure 3 were not clear. Figures 3 and 4 and Section 3 are AERONET data. We rephrase the caption of Figure 3a and 3b: "Multi-year mean seasonal cycle of observed AERONET $AOD_{550nm}$ at long-term sites", and add "observed…from AERONET" to the captions of Figure 4a and 4b. We further clarify the title of Section 3 (added text underlined): "Climatology of AERONET AOD and $\alpha_{ext}$ observations over Africa…" and add "AERONET AOD and $\alpha_{ext}$ observations" to the title of subsections 3.1, 3.2, and 3.3. We also explicitly state more frequently throughout Sections 3 and 4 when we are referring to AERONET AOD observations vs. model results.

Page 14, line 5: As another style example, "The Pearson's correlation coefficient" could have "The" and probably "Pearson's" deleted as well.

We implement this suggestion (now page 14, line 6).

Page 14, line 16: is the beta here intentional? If so, what does it mean?

This was a typo and has been removed – thank you for catching it.

Figure 3: In general I don't see the point of these figures. Seeing one line per site here is not very informative. If the purpose of the paper is to compare with AERONET, the same basic information for AOD is repeated in Figure 6. Or am I misunderstanding something? It would be better to show, for each site, the model and AERONET together so a direct comparison can be made. So something like Figure 6, for both AOD and AE.

As stated earlier, it appears there was a misunderstanding of Section 3 and Figures 3 and 4, which we clarified in Section 3, the Figure captions, as well as Section 4 following your helpful comments. Figures 3 and 4 present the observations only. We felt this complete record of observed AOD at sites influenced by African dust and biomass burning could stand on its own outside of the model evaluation. Therefore, we include more sites in these figures even though they have limited data coverage - e.g., not a full seasonal cycle, or only a single year of observations - which make them not very useful for evaluating the climate model but still informative to get the broader picture of observed AOD across the African continent and outflow regions. We explain in section 2.2, page 6 lines 8-11, how we selected observational sites with which to evaluate the model, and discuss the temporal resolution limitations of modeled emissions of aerosols and their precursors from CMIP5 in section 2.1, page 5 lines 9 to 16.

We clarify Figure 6 within Section 4.2 (now page 11, line 15): "Figure 6 shows the same multi-year mean seasonal cycle for observed AERONET AOD as in Fig. 3 (here in red triangles)…". We think this and the clarifications made within Section 3 and to the captions of Figures 3 and 4 described earlier should address your comment.

Modeled AE is not possible to obtain. Modeled AOD is only calculated at 550nm.

Figure 7: It would be better to overplot the AERONET AOD on top of the model component lines, rather than shifting it off to the right, to allow a more clear visual comparison of aerosol amount and seasonality.

We edit Figure 7 following your suggestions.

Table 1: It would be useful to perform the AERONET/model comparison at ALL the sites shown, not just a subset. Otherwise what is the point of including them in the paper if the AERONET data are not used?

See response to previous comment above regarding Figure 3, and earlier comment regarding Section 3.

Table 3: I am not sure it is useful to report significance of correlation coefficients here. I don't think that it adds anything to the analysis or discussion, and due to strong autocorrelation of the data (which I don't think is accounted for) it is possible that the significance estimates are incorrect anyway.

We have performed an autocorrelation test on the model output; the observational dataset had many missing monthly averaged values, and thus the analysis was performed on the model output that has a complete dataset. In order to ensure the annual cycle does not have a role in the autocorrelation analysis, we performed the analysis per month per site (e.g., assessing autocorrelation in all January means for Skukuza site). As this analysis was per site and per month, the n=360. At a time lag = -1, there were only 6 instances out of these 360 (1.6%) where the autocorrelation was statistically significant at a 95% confidence interval. This is a very small fraction of the data analysed, and thus the autocorrelation in the model output can be considered not statistically significant. As no autocorrelation was found in the model output, it is assumed that there is not autocorrelation in the observed data as well.

General: as noted in my Quick Report, I suggest the authors also perform some analysis using daily (rather than monthly) data. This can be simple visual scatter plots for each site, or something similar to Table 3. This will help to tell to what extent biases in the monthly data are due to aerosol events that are missed in the model, and to what extent they are systematic biases in component loadings or optical properties. Going to daily data here also helps to avoid some of the sampling differences.

We add a comparison of daily data (now Figure 8) and corresponding discussion (new Section 4.2.4). We introduce the daily comparison at the end of Section 2.3 (Page 6, lines 5-10): "We also compare modeled daily average $AOD_{550nm}$, using the same daylight hours previously described, to observed AERONET daily average $AOD_{550nm}$ for the specific days with available data at each site. As described in Section 2.1, outside of the dust parameterization, the experimental setup of the model following CMIP5 does not take daily variations in emissions into account, and thus the daily variation in modeled AOD from all other aerosol types will be due to daily variations in transport and removal only. Even with these limitations, the daily comparison is useful for further investigating model biases."

---

## Author Comment (AC2) · 12 Sep 2017

Thank you for your thoughtful comments. Including your suggested revisions has improved the quality of the manuscript. Our responses are indicated below in blue text.

Reviewer 2

Interactive comment on

"Understanding the seasonality and climatology of aerosols in Africa through evaluation of CCAM aerosol simulations against AERONET measurements"

Anonymous Referee #2

General comments: In this study the regional and seasonal representation of aerosols in the global CCAM model is evaluated for the African domain, mainly through comparison of modeled and ground based retrievals of AOD from AERONET across Africa and parts of the Middle East. I find the paper scientifically interesting and mostly well written, and the presentation of model and observationally based results should be useful for others planning to do similar model evaluation studies. Parts of the model description is vague, however, which makes it difficult for the reader to find necessary information about the aerosol treatment without actually reading many of the underlying papers for the model. The treatment of sea salt, in particular, is poorly described, and the potential impact of this component (on coastal and island sites) on the results has mainly been omitted, except (for some sites) in Figure 7.

We thank the reviewer for their input. We have expanded the description of the model and its treatment of sea salt (specific additions are detailed below). In the model, sea salt is not transported, and thus over land there is little influence on the simulated aerosol properties. However, observations at some sites could be impacted by sea salt and the manuscript has been updated to acknowledge this (Section 3). Our responses to the points below are in blue.

Specific comments and technical corrections: (For simplicity, the arrow symbol "->" is used to suggest a change from text version A -> B)

Page 1, line 18: "ground-based observations" should be changed to "ground-based remote retrievals" or something along that line.

This has been changed as suggested to "ground-based remote retrievals" (now page 1, line 17).

Page 3, line 8: "may also feedback on climate" -> "may also feed back on climate" (feedback is a noun)

This has been updated to "may also have a feedback on climate" (now page 2, line 33).

Page 3, line 15: "first AeroCom" -> "first phase of AeroCom"

This has been updated to "first phase of AeroCom" (now page 3, line 7).

Page 3, line 30 and throughout Sect. 2.1: What does non-prognostic / diagnostic sea salt aerosols mean? Do you prescribe the emissions or the concentrations? A reference for this treatment should be added, e.g. after the additional (but not sufficient) info on page 5, line 3.

Additional clarification is provided in Section 2.1, which was updated to include the following (now page 4, lines 10-15): "Sea salt concentrations above the ocean surface are diagnosed (i.e. prescribed) at each time step as a function of the 10-m wind speed. It is assumed that sea salt aerosols are well-mixed in the marine boundary layer (MBL), and that the concentration is zero above the MBL. There are two size bins of sea salt aerosols (mode radii of 0.035 μm and 0.35μm). As the sea salt concentrations are prescribed at each time step, they are not actively emitted, transported or removed, and thus no sea salt is transported over land (Rotstayn et al., 2007)."

Page 4, line 12; "spun-up" -> "spun up"

This has been corrected.

Page 4, line 19: Is also the semi-direct effect taken into account?

This has been clarified, the following text has been added (page 4, lines 20-22): "The semi-direct effect is also included in CCAM; however, as the vertical temperatures upwards of 900 hPa are nudged towards the ERA-Interim reanalysis data every six hours in accordance with CORDEX, the semi-direct impact on the simulation presented here is diminished."

Page 4, line 31: The "-" in "-2" in the exponent (m**-2) is misplaced.

Page 5, line 13: "vary every 5 years" "they vary every 5 years"

Page 5, line 14: "anthropogenic" -> "non-biomass burning anthropogenic"

Page 5, line 17: "found a chemical" -> "found that a chemical"

The above four corrections have been made in the text as suggested.

Page 5, line 17: Unless sea salt concentrations are prescribed, why are these large particles not also affected by gravitational settling?

The sea salt concentrations are prescribed. Clarification was added as described in the response to an earlier comment in Section 2.1.

Page 6, line 2: "AOD" -> "AOD at 550 nm" Page 6, line 5: "34 sites Africa" -> "34 sites in Africa"

Page 6, line 18: "bolded site names" -> "site names in bold font"

The above two corrections and suggestions have been adopted in the text as written. We also similarly revised the caption of Tables 1 and 2.

Page 6, lines 24-25: I would suggest to rewrite "where if more than 30% of the daily values were missing, a monthly average could not be calculated " to ". I.e., if more than 30% of the daily values were missing, a monthly average was not calculated ".

This has been corrected to: "(i.e., if more than 30% of the daily values were missing, a monthly average was not calculated for that time period)" (now page 6, lines 17-18).

Page 6. line 27: "This is to ensure the" -> "This is to ensure that the"

This has been edited as suggested above.

Page 6, lines 30 and 32: The sentence containing "and were aligned as possible" does not make sense, and the meaning of the following sentence is not clear to me either: Should it read "assessed for the averaging period in question" or "assessed for the respective averaging period"?

This sentence has been removed and the entire paragraph clarified following your and the other reviewers' suggestions, as follows (page 6, lines 23-28): " Daily average AOD from AERONET is calculated for a minimum of 3 time points from sun photometer measurements, which can only be made during daytime, while modeled AOD is reported at 6-hourly resolution. Therefore, only CCAM AOD between 06:00 and 18:00 UTC was averaged for monthly and multi-year means (similar to other AERONET-model comparison studies; (e.g., Tegen et al., 2013). Model monthly means were, however, insensitive to the choice of daylight cut-off (see Fig. 2), which gives confidence that the instantaneous 6-hourly values from CCAM can represent the range of daytime hours sampled by AERONET."

Page 7, lines 5-9: Unclear description of the 2 different calculations: "2) all model years" does not preclude 1). Should it be "2) all months of all model years"?

This has been updated to "all months of all model years".

Page 7: Eqs. 4 and 5 are well known and can be skipped, or replaced with an equation for r. Line 17 also repeats the info on line 15.

The two equations have been deleted from the main text and moved to a footnote of what is now Table 4 (formerly, Table 3).

Page 7, line 25: The Angstrom parameter does not equal 2 for all sub-micron particles. It is more correct to write "very fine particles" or "predominantly fine particles", or something along that line.

This sentence has been updated following this suggestion and that of the other reviewer (now page 7, lines 16-18): "The Ångström exponent is an empirical proxy related to the relative contribution to optical thickness from coarse vs. fine aerosols, with values varying between approximately 0 for pure coarse dust particles to 2 for predominantly fine particles (Leon et al., 2009; Hamonou et al., 1999)."

Page 7, line 27 (and throughout the manuscript): Small Angstrom parameter values can also be due to aerosols dominated by coarse sea salt aerosols. Perhaps this is not the case for this particular model and the sites studied here, but this should somehow be shown, at least for the coastal and island sites.

This section is referring to AERONET observations, and thus this has been clarified to, "…indicative of aerosols dominated by coarse particles (e.g., mineral dust or coarse sea salt particles)…". (Page 7, line 24). We update the caption of Figure 4 to be consistent.

We add the following to section 3.1 (now page 8, lines 8-11): "While low values of $\alpha_{ext}$ could represent other coarse particles besides dust like sea salt, previous work has indicated sea salt is a minor contributor to aerosols at island sites to the north of Africa, including Izaña (Rodríguez et al., 2011; Putaud et al., 2000; Querol et al., 2009). The correspondence of the seasonality in $\alpha_{ext}$ and AOD with known dust events suggests mineral dust is the primary contributor to extinction from coarse particles."

See previous comments for discussion of modeled sea salt.

Page 9, line 4: As above.

We change "coarse dust aerosol" to " coarse aerosol particles, most likely dust" and then add the following discussion (page, lines 4-9): ". Previous work found that minimum values of $\alpha_{ext}$ are related to dust storms at Ouagadougou, Dakar, and Agoufou, and clearly linked to dust at Ilorin and Banizoumbou based on air mass back trajectories and observed seasonality (Ogunjobi et al., 2008). While Dakar is frequently influenced by air transported over the Atlantic Ocean (Ogunjobi et al., 2008), analysis off the coast of Dakar at Cape Verde found the AOD and aerosol mass loading were dominated by desert dust, with sea salt minimally contributing to AOD (6%) in part due to its small extinction (Chiapello et al., 1999) which would also imply a minor influence on $\alpha_{ext}$.

Page 10, line 31: "fraction wet deposition" -> "wet deposition fraction"

This has been updated to "wet deposition fraction"

Pages 10-11: Whether this aerosol is prescribed or not, sea salt influences the total AOD and should be discussed and included in Table 2, and also in Fig.5 if prescribed concentrations have not been used.

The description of prescribed sea salt has been updated (Section 2.1, now page 4, lines 10-15) as described in a previous response, which may address this comment. Of the information presented in Table 2 and Figure 5, only the burden of sea salt could be computed, but this would not be very meaningful given concentrations are prescribed only within the mean boundary layer and set to zero everywhere else. Thus sea salt has not been included in Table 2 and Figure 5.

The sea salt AOD values for coastal/island sites in Fig. 7 look small compared to many of the available AerCom models (http://aerocom.met.no/cgi-bin/aerocom/surfobs_annualrs.pl).

We add a sentence to page 15, lines 16 to 18: "A small impact of simulated sea salt can be seen at the Santa Cruz Tenerife site (Fig. 7) (mean AOD of 0.04). The sea salt contribution to simulated monthly AOD at 550nm from AeroCom Phase III-CTRL2015 (AeroCom Phase II Interface, 2017) ranges from negligible to greater than 0.1 at Santa Cruz Tenerife."

Page 11, line 1: "BC are higher" -> "BC burden and lifetime are higher".

This has been updated to "BC burden and lifetime are higher".

Page 11, line 23: "areas are ±1 standard deviation" -> "areas are within ± 1 standard deviation" (and the same for the following line).

"within" has been added to both lines.

Page 12, line 1: "spurious summertime peaks" -> "missing summertime peaks"

This has been updated to "missing summertime peaks".

Page 12, lines 11-12: Since there is a severe overestimate in modeled AOD for some sites and some months, the sentence starting with "The model generally represents the magnitude of AOD550nm" should be rephrased.

We revise the sentence (now page 12, lines 12-13): " In comparison to the other regions, the model better represents the magnitude of AOD at the southern African sites (except for Reunion Island) with a smaller normalized mean bias and mean absolute error (see Fig. 6 and Table 4)."

Page 13, line 8: "The AOD550" -> "The observed AOD550" (?)

This has been updated to "The observed AOD".

Page 13, line 15: "at the source" -> "at the biomass burning source"

This has been updated to "at the biomass burning source".

Page 14, line 1:  Is the ERA wind bias in winter consistent with the magnitude of the AOD bias? Could you make a simple estimate of this?

We are currently exploring the reasons which may contribute to the AOD bias; however, this is a complex problem that falls beyond the scope of this paper.

Page 14, line 16: There is a "beta" to much in "0.27 m s-1ß"

This was a typo and has been removed – thank you for catching it.

Page 14, line 31 and Page 16, lines 17-20: Can you show that the precipitation in the model is underestimated compared to observations (therefore explaining some of the positive dust emission bias)?

The model does not exhibit a significant dry bias in the Sahel and Sahara. This analysis with model set-up from this manuscript is not published; published results of a previous version of model that highlight the wet bias of the model in representing the average daily summer rainfall totals over most of Southern and Tropical Africa (Engelbrecht et al., 2011), Therefore, we suspect that the overestimation of dust is the result of an overestimation of near-surface wind speeds and/or source regions. However, this analysis is ongoing.

Reference: Engelbrecht FA et al., 2011, Multi-scale climate modelling over South Africa using a variable-resolution global model, Water SA, 37(5).

Page 16, line 31: "is slightly underestimated" -> "is underestimated"

This has been updated to "is underestimated".

Page 17, line 20: "CCAM is able to capture the general seasonal cycle of the emissions of dust, and the transport of all aerosol types". This has not been shown, and such a statement should be limited to the aerosol components covered by the study.

This has been updated to "…transport of dust, carbonaceous and sulfate aerosol types."

Table 1: The second sentence in the table caption is grammatically incomplete.

The previous Table 1 has been split up to Table 1 and Table 2. The heading has been changed and is now complete. "Table 1: AERONET site information (site names in bold font indicate those sites used in model comparison). The average (± 1 standard deviation) and median (25th and 75th percentile) values for $AOD_{550}$ and $\alpha_{440-870}$ per site are shown."

Table 2: The gray shading should be explained in the table caption (as in the text).

The gray shading is to distinguish the different regions studied. A column has been added to identify these regions (as in Table 1). Due to the addition of a new table, this is now Table 4.

Figure 1, caption: "used in model comparison" -> "used in the model comparison".

This has been updated to "used in the model comparison".

Figures 2-4, caption: Explain the whiskers.

To the caption of Figure 2, we add: "Whiskers are ±1 standard deviation across the 6-hourly model values within each time range." "Whiskers are ±1 standard deviation" is added to the captions of Figures 3 and 4.

---

## Author Response (AR2)

Thank you again for your comments. Our responses are indicated below in blue text.

I reviewed a previous version of this paper. In general, the authors have addressed my concerns, and I commend them for their efforts in the revision. I have a small number of remaining comments, but after that feel the paper can be accepted for publication in ACP. I am happy to read a future revision if the Editor would like, but I don't feel that it is a necessity. The comments mostly relate to the presentation of some of the Figures, where what is shown is not clear from what is written.

General: I had suggested adding an evaluation of model Ångström exponent to the paper. The authors respond that the model does not provide this, only the AOD at 550 nm. That's fine, but I suggest noting this clearly somewhere up front (like at the end of the introduction and/or the start of section 2.3). Otherwise a reader may wonder if you chose to 'hide' model Ångström exponent or something.

we add the following text in the first paragraph of Section 2.3: " Modeled Ångstrom exponent is not available."

Page 3 line 20 and page 5, line 25: the authors cite both Holben et al (1998) and Dubovik et al (2002) for the AERONET network here. The Holben paper is the correct reference for the network as a whole. Dubovik presents some climatological results for AERONET's aerosol inversions, which are not the data set the authors use (they use the direct Sun data). So this is not really relevant. I suggest the authors cite only Holben et al (1998) in these two places.

We remove the Dubovik et al. citations.

Page 7, lines 18-19: "The Ångström exponent values presented here are based on aerosol extinction." I suggest deleting this sentence. It seems out of place, and anyway this is the standard definition which was already given in Equation 2.

We remove this sentence.

Figures 3 and 4: I understand now that this is AERONET data, not model simulations. If this is the seasonal cycle and multi-annual means and standard deviations, then why are there standard deviations for sites and months where only 1 year contributed? For example in KAUST there are data for March-June, the numbers above the points indicate 1 year contributed to each month in the seasonal cycle, but there are large standard deviations. If there is only 1 year contributing then the standard deviation can't be calculated, by its definition, because you only have 1 data point going into the mean. Is this an error or is this not in fact the mean seasonal cycle? Perhaps it is calculated as a mean of instantaneous data or of daily means instead? In that case the standard deviations would include both day to day and year to year variability (which is not what is implied). Either the plot needs correcting or the caption updating to reflect what is shown. My preference would be to take the AERONET daily product and average it to monthly means. Then, calculate the annual cycle (multi-year mean and year-to-year standard deviation) based on the monthly mean values. In this case sites like KAUST will end up without standard deviations, for that reason. But the current presentation of the figure is not consistent with what it says. Or have I totally misunderstood?

We revise the figure captions in Figures 3, 4, and 6 to include additional text, " across daily means within a given month", describing the standard deviation. This is the standard

deviation across daily means within a given month for all years meeting the data completeness cutoff. We represent the variability this way because we do not expect the model to be able to capture individual days, and hence feel this is the appropriate level of variability to compare the model against. We also only include individual months with greater than 70% of days having valid daily mean AERONET AOD observations.

Figure 5: this appears before Figures 3 and 4 in the manuscript file uploaded (acp-2017-250-manuscript-version5.pdf).

This was an error introduced in the process of combining all figures into a single PDF, thank you for catching this.

Figure 6: the same comment about averaging for Figures 3 and 4 may apply here, it is not clear from the description.

See comment above – we also revise this figure caption.

Figure 9: it would be good to have the top and bottom rows on the same scale (top is 0-1.5 now, bottom is 0-1) to more easily compare the two time periods. Maybe putting both on the range 0-1.2 is a good compromise?

We put both Figure 9a and Figure 9b in the same range as suggested.

[revised manuscript text omitted]

a) March

| 5th percentile | Mean | 95th percentile |

b) September

| 5th percentile | Mean | 95th percentile |

[Figure]

**Figure 9: Seasonal variation in multi-year monthly 5th percentile, mean, and 95th percentile modeled AOD$_{550nm}$ (map background) for the full model climatology (1999–2012), with observed multi-year means (points) for all available AERONET data, for a) September and b) March. The number of years and range of years used for each site is the same as in Fig. 3.**